# Camsap2a regulates actomyosin flow and Rab5ab-mediated macropinocytosis in the yolk cell during zebrafish epiboly

Haoyu Wan*, Sifa Quibria*, Ernest Iu, Sirma Damla User, Sergey V. Plotnikov and Ashley E. E. Bruce‡

## ABSTRACT

In zebrafish, epiboly is a major morphogenic event during gastrulation, characterized by the thinning and spreading of the embryonic blastoderm to internalize the underlying extra-embryonic yolk cell. This movement is driven by the yolk cell, which generates motile force through actomyosin flow that engages a circumferential contractile band, pulling the attached blastoderm vegetally. Localized macropinocytosis of the yolk cell, another actin-driven process, also contributes to epiboly progression by removing yolk membrane ahead of the advancing blastoderm. The molecular mechanisms coordinating these processes are elusive. Here, we identified Camsap2a, a non-centrosomal, microtubule-stabilizing protein, as a regulator of actin-dependent processes in the yolk cell during epiboly. Epiboly is delayed in *camsap2a* mutant embryos, which exhibit reduced macropinocytosis as well as impaired actin flow, contractile band formation and function. We show that Camsap2a functions in actin regulation upstream of the small GTPase Rab5ab, as constitutively active Rab5ab rescues the defects in macropinocytosis, actomyosin band formation and epiboly. Our work provides new insights into the molecular control of epiboly and further implicates membrane dynamics as an important contributor to the process.

KEY WORDS: Zebrafish, Epiboly, Gastrulation, Morphogenesis, Actomyosin, Macropinocytosis, Camsap, Rab5

## INTRODUCTION

During gastrulation, germ layers are specified and positioned to generate the adult body plan. Before gastrulation begins, the late blastula-stage zebrafish embryo comprises a multi-layered blastoderm that generates all tissues and sits on top of a large extra-embryonic yolk cell. The blastoderm consists of an outer epithelium, the enveloping layer (EVL) and inner deep cells, while the yolk cell contains syncytial nuclei in the yolk syncytial layer (YSL) (Kimmel et al., 1995). The first coordinated morphogenetic event is epiboly when the EVL, deep cells and YSL spread towards the vegetal pole to enclose the yolk cell as an essential step in generating the adult body plan (Kimmel et al., 1995; Warga and Kimmel, 1990).

The external yolk syncytial layer (e-YSL), positioned adjacent to the blastoderm margin, provides the major motile force for teleost epiboly (Fig. S1C) (Behrndt et al., 2012; Köppen et al., 2006; Trinkaus, 1951). Two contractile mechanisms in the yolk cell e-YSL have been described: a circumferential motor and a flow-friction motor produced by the animally directed actomyosin flow (Behrndt et al., 2012). Starting at late blastula, actin and myosin flow from the vegetal pole towards the margin and are assembled into a circumferential actomyosin band in the e-YSL (Behrndt et al., 2012). After the blastoderm spreads past the equator of the yolk cell, the actomyosin band begins to contract circumferentially to pull the mechanically linked EVL to the vegetal pole, while EVL cells and the deep cells rearrange to allow the blastoderm to spread. As the blastoderm expands vegetally, yolk membrane in front of the advancing margin is removed by macropinocytosis, a process that also depends on actin in the e-YSL (Betchaku and Trinkaus, 1986; Cheng et al., 2023; Solnica-Krezel and Driever, 1994).

In addition to actin, there is an extensive network of microtubules in the yolk cell, including a longitudinal array along the animal-vegetal axis associated with the external yolk syncytial nuclei (e-YSN). Before epiboly begins, the YSN become postmitotic and our previous work suggested that, as commonly seen in postmitotic cells, the longitudinal microtubule array might become non-centrosomal at this time (Fei et al., 2019). Disrupting yolk microtubules using microtubule-stabilizing or -depolymerizing agents results in moderate epiboly delays (Jesuthasan and Strähle, 1997; Solnica-Krezel and Driever, 1994), suggesting that microtubules might play a role in facilitating the formation or function of the actomyosin ring. This led us to consider a potential role for the calmodulin-regulated spectrin-associated (Camsap) family of microtubule minus-end-binding proteins in regulating the yolk cytoskeleton during epiboly.

Camsap proteins are evolutionarily conserved in metazoans and are implicated in stabilizing non-centrosomal microtubule minus ends and mediating microtubule–actin interactions in mammalian tissue culture and invertebrate models (Baines et al., 2009; Jiang et al., 2014). Structurally, they consist of an N-terminal Calponin homology (CH) domain, which has been postulated to be descended from a protozoan CH2 domain (Baines et al., 2009). In the middle of the protein, there are three coiled-coil domains (CC1-CC3), which are associated with microtubule minus-end recognition and binding (Jiang et al., 2014). Work in mammalian cell culture showed that the CC1 region can promote actin–microtubule crosslinking during cell migration (Boldt et al., 2016; Ning et al., 2016; Go et al., 2021). The C terminus of Camsap proteins contains the defining domain of the protein family, the CAMSAP1, KIAA1078 and KIAA1543 (CKK) domain, which binds to and stabilizes non-centrosomal microtubule minus ends (Atherton et al., 2017; Gong et al., 2018; Hendershott and Vale, 2014; Ho et al., 2023; Jiang et al., 2014). Camsaps can

Department of Cell and Systems Biology, University of Toronto, Toronto, ON M5S 3G5, Canada.
*These authors contributed equally to this work

‡Author for correspondence (ashley.bruce@utoronto.ca)

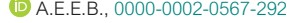 A.E.E.B., 0000-0002-0567-2928

also affect actin dynamics and vesicle trafficking; in *Caenorhabditis elegans*, the CH domain of the Camsap homolog PTRN-1 activates formin, which stimulates actin polymerization during endocytic recycling in the intestinal epithelia (Gong et al., 2018).

We considered a potential role for the Camsap family in regulating the yolk cytoskeleton during epiboly. Relatively little is known about the developmental functions of Camsaps but *camsap2a* (one of five Camsap genes in zebrafish) was previously shown to be restricted to the YSL during gastrulation (Hong et al., 2010; Sprague et al., 2001). We used CRISPR/Cas9 gene editing to generate two *camsap2a* mutant alleles that both produce embryos with epiboly delays. Surprisingly, we found that yolk microtubules appeared relatively normal in mutant embryos. In contrast, actomyosin accumulation to form the contractile ring was delayed and reduced in mutant embryos. In addition, yolk membrane removal by macropinocytosis was reduced in mutant embryos. The *camsap2a* mutant phenotype closely resembled that induced by yolk-specific morpholino knockdown of Rab5ab, a small GTPase (Kenyon et al., 2015; Marsal et al., 2021), and we found that macropinocytosis, actin accumulation and the overall epiboly delay in *camsap2a* mutant embryos could be restored by expression of constitutively active Rab5ab. Our findings suggest that the primary role of Camsap2a in the yolk cell is to regulate the activity of Rab5ab.

## RESULTS
### Maternal-zygotic *camsap2a* mutant embryos exhibit delayed epiboly progression
To identify regulators of the yolk cell cytoskeleton during zebrafish gastrulation, we focused on *camsap2a* because its transcript was previously shown to be restricted to the YSL during gastrulation, which we confirmed (Fig. 1A) (Hong et al., 2010). To investigate Camsap2a function, we used CRISPR/Cas9 and a guide RNA targeting exon 5, which encodes the N-terminal calponin-homology (CH) domain to knock out all the functional domains of the protein (Fig. 1B). Two mutant alleles were recovered: *camsap2a*[uot19] and *camsap2a*[uot20]. The *camsap2a*[uot19] allele is a 12 bp in-frame deletion, predicted to produce a 4-amino-acid deletion in the CH

domain while *camsap2a*[uot20] is a 13 bp deletion in the CH domain that results in a premature stop codon (Fig. 1B).

Although RNA-sequencing data indicated that *camsap2a* is zygotically expressed starting at the blastula stage (White et al., 2017), we successfully amplified *camsap2a* from cleavage-stage cDNA (Fig. S1A), indicating that the transcript is also present maternally. Our initial experiments in-crossing heterozygous fish produced zygotic mutant embryos that displayed mild epiboly delays with low penetrance (data not shown); therefore, we generated maternal-zygotic (MZ) *camsap2a* mutant embryos from homozygous mutant fish.

Both MZ*camsap2a*[uot20] and MZ*camsap2a*[uot19] mutant embryos exhibited epiboly delays, which were most apparent after shield stage (6 h post-fertilization, hpf), while epiboly initiation was largely unaffected (Fig. 1C-F). When wild-type embryos were at 8 hpf (75% epiboly), MZ*camsap2a*[uot20] embryos were around 60% epiboly (equivalent to 7 hpf) (Fig. 1D), and a similar delay was seen in MZ*camsap2a*[uot19] embryos (Fig. 1F). We observed that some embryos from both alleles were abnormally elongated along the animal-vegetal axis (Fig. 1D,F) and quantification of embryo shape in MZ*camsap2a*[uot20] embryos confirmed that they were less round than wild-type embryos at 8 hpf (Fig. S1B). Despite the difference in the embryo shape, the MZ*camsap2a* mutants were able to complete epiboly and could be raised to fertile adults. Nevertheless, our data demonstrated a prominent (1-1.5 h) delay in epiboly of MZ*camsap2a* mutant embryos.

We examined the transcript levels of all five Camsap genes by quantitative PCR. We focused on MZ*camsap2a*[uot20] mutants because they contain an early stop codon predicted to lead to nonsense-mediated decay of *campsa2a* and upregulation of homologous genes by transcriptional adaptation (Falcucci et al., 2025; Rossi et al., 2015; Sztal and Stainier, 2020). The results showed that *camsap1a*, *camsap1b*, *camsap2b* and *camsap3* transcript levels were upregulated in mutant embryos compared to wild type (Fig. S1D). The *camsap2a* transcript was not decreased compared to wild-type embryos, which could be because the mechanisms or machinery for nonsense-mediated decay function differently in the yolk cell or a regulatory mechanism is triggered in

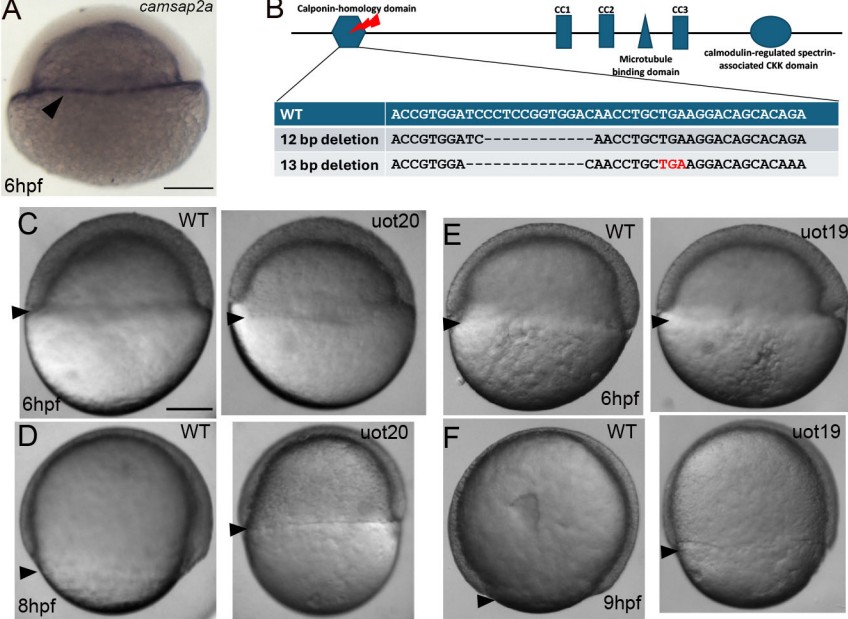

**Fig. 1. Epiboly is delayed in MZ*camsap2a* mutant embryos.** (A) Whole-mount *in situ* hybridization for *camsap2a* at 6 hpf (shield stage). Arrowhead indicates transcript localization in e-YSL. Lateral view. Scale bar: 150 µm. (B) Schematic of the Camsap2a protein. Red bolt indicates CH domain, which was targeted by CRISPR/Cas9 gene editing. Enlarged region shows genomic sequence of the two mutations generated: a 12 bp in-frame deletion (*camsap2a*[uot19]) and a 13 bp nonsense mutation (*camsap2a*[uot20]). Stop codon is shown in red. (C-F) Bright-field images of wild-type, MZ*camsap2a*[uot20] and MZ*camsap2a*[uot19] embryos at 6 hpf (C,E) and 8-9 hpf (D,F). Arrowheads indicate the margin, and extent of epiboly. Lateral views. Scale bar: 150 µm. WT, wild type.

mutant embryos that leads to upregulation of *camsap2a* transcription. While the upregulation of the other Camsap genes likely explains the viability of the mutant embryos, the robust rescue effect of full-length Camsap2a (described below) demonstrates that they cannot fully compensate for the Camsap2a mutation.

## Yolk microtubule organization appears normal in MZ*camsap2a* mutant embryos

To characterize the mutant phenotype, we first examined the yolk cell microtubule cytoskeleton. Based on the known functions of Camsap proteins, we expected that there could be prominent changes in microtubule abundance, organization or dynamics (Goodwin and Vale, 2010; Hendershott and Vale, 2014; Richardson et al., 2014; Tanaka et al., 2012). We examined microtubules by immunohistochemistry for alpha-tubulin and did not observe changes in microtubule organization in mutant embryos compared to wild-type embryos (Fig. 2A). Notably,

in wild type and mutants of both alleles, we saw yolk syncytial nuclei in the process of migrating towards the vegetal pole, which requires an intact microtubule network (Fig. 2A, arrowheads) (Fei et al., 2019).

We then examined the structure and dynamics of yolk microtubules in live embryos. RNA encoding the microtubule-binding domain of ensconsin fused to three GFP molecules (EMTB-3XGFP) (Miller and Bement, 2009) was injected into one-cell-stage wild-type and mutant embryos, and embryos were imaged by time-lapse confocal microscopy starting at mid-epiboly stages. Yolk microtubules were present in mutant embryos and the organization of the microtubule array in the yolk cytoplasmic layer appeared normal in both mutants compared to wild-type embryos (Fig. 2B-D). MZ*camsap2a* mutant embryos expressing EMTB-3XGFP did not show obvious changes in microtubule abundance in mutant compared to wild-type embryos; however, qualitatively, microtubules in MZ*camsap2a*[uot19] embryos looked slightly disorganized compared to controls (Fig. 2C). Overall,

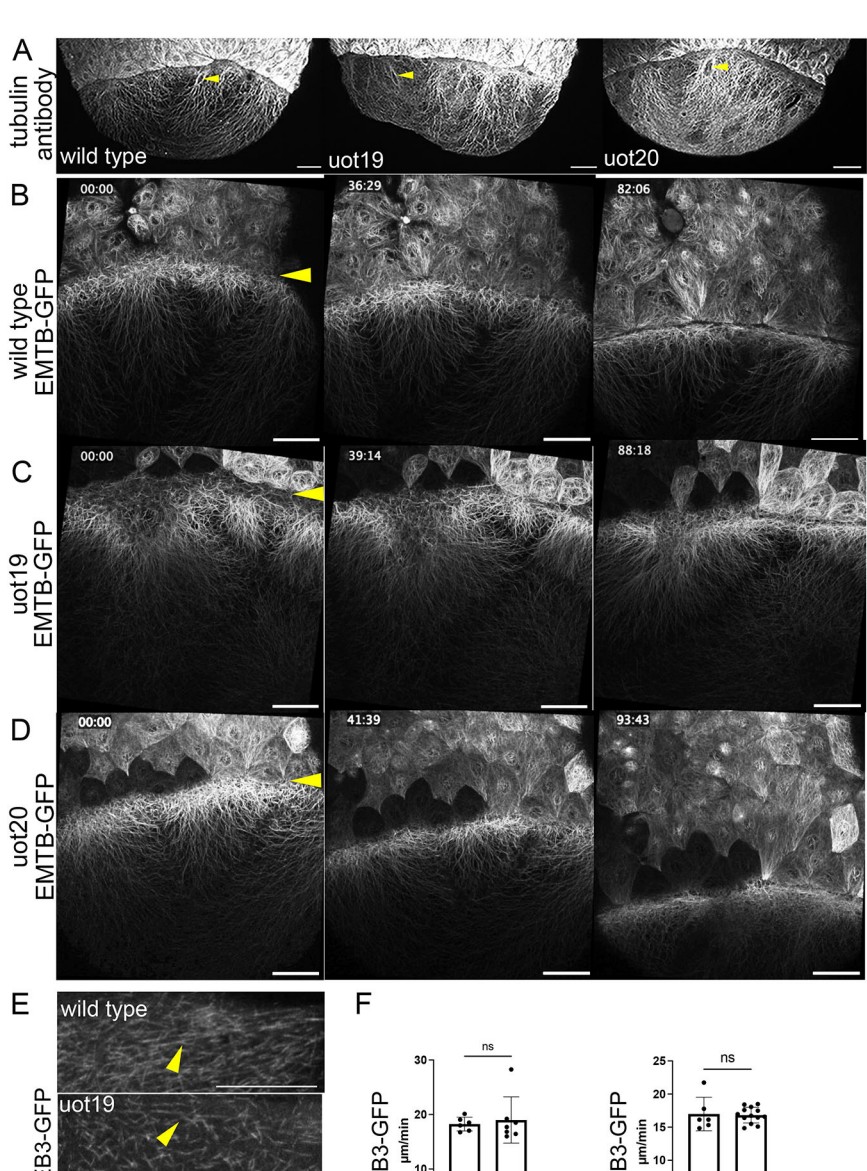

**Fig. 2. Microtubule defects are not detected in MZ*camsap2a* mutant embryos.** (A) Alpha-tubulin antibody staining in wild-type, MZ*camsap2a*[uot19] (uot19) and MZ*camsap2a*[uot20] (uot20) embryos at 7 hpf (60-70% epiboly). Arrowheads indicate unlabeled external yolk syncytial nuclei surrounded by microtubules. (B-D) Stills from confocal time-lapse movies of EMTB-3XGFP-labeled microtubules in wild-type (B), uot19 (C) and uot20 (D). Arrowheads indicate the yolk-blastoderm boundary. Imaging started at 6 hpf (shield). Time stamp indictates min:s. Scale bars: 50 μm. (E) EB3-GFP in the e-YSL of wild-type embryos at 7 hpf and time-matched mutant embryos. Arrowheads indicate horizontal EB3-GFP. Scale bars: 30 μm. (F) Quantification of e-YSL EB3-GFP speed in wild-type (*n*=6) and mutant embryos (uot19, *n*=7; uot20, *n*=13). Data are mean±s.e.m. *P*=0.8792 (left), *P*=0.6702 (right) (Welch's *t*-test). ns, not significant; WT, wild type.

these results indicated that yolk microtubule organization was not strongly affected in MZ*camsap2a* mutant embryos.

We next investigated the effects of MZ*camsap2a* mutations on microtubule dynamics using EB3-GFP, which binds to polymerizing microtubule plus ends and was previously used to assess yolk microtubules during epiboly (Eckerle et al., 2018). Examination of EB3-GFP-expressing wild-type and mutant embryos by spinning disk confocal microscopy did not reveal obvious differences in EB3 directionality (Fig. 2E). Quantification of EB3 comet speed (Applegate et al., 2011) also did not show significant changes between wild-type and mutant embryos (Fig. 2F). Overall, yolk microtubule abundance and polymerization rates were not significantly altered in MZ*camsap2a* embryos, consistent with the conclusion that microtubule defects are unlikely to be a major contributor to the mutant phenotype. To further investigate the cause of the epiboly delay in mutant embryos, we examined the yolk actin cytoskeleton.

### Yolk actin accumulation is reduced in MZ*camsap2a* mutant embryos

At the onset of epiboly, yolk actin and myosin flow from the vegetal pole and accumulate in the e-YSL where they form the contractile ring that provides the major motive force for epiboly (Behrndt et al., 2012; Hernández-Vega et al., 2017; Köppen et al., 2006). To determine whether actomyosin ring formation or function were altered in MZ*camsap2a* mutant embryos, we examined F-actin by phalloidin staining. Confocal imaging and quantification of fluorescence intensity in wild-type embryos at 8 hpf (75% epiboly) and time-matched MZ*camsap2a* mutant embryos at 60% epiboly showed that e-YSL actin accumulation was reduced in embryos from both mutant alleles (Fig. 3A,B; Fig. S1C). Given that most MZ*camsap2a* mutant embryos finished epiboly, we hypothesized that e-YSL actin accumulation might recover at later time points, thereby enabling the completion of epiboly. To test this possibility, wild-type and stage-matched mutant embryos were fixed at 75% epiboly and stained with phalloidin (Fig. 3C). Actin fluorescence intensity measurements showed that actin accumulation in mutant embryos recovered at later stages by comparing stage-matched mutant embryos to time-matched mutant embryos (Fig. 3D, see Materials and Methods). We observed reduced EVL actin

fluorescence intensity in MZ*camsap2a* mutant embryos, which we suspected was secondary to the reduction in pulling force from the yolk actomyosin ring (Marsal et al., 2021) and which was confirmed by the rescue experiments described below.

### Camsap2a regulates yolk actin accumulation by altering actin flow

We postulated that the reduced actin accumulation in the e-YSL in mutant embryos could result from abnormal flow and/or altered actin dynamics. To investigate whether the actin defects in mutant embryos were due to abnormal flow, actin was imaged using spinning disk confocal microscopy. Short (3-min) time-lapse movies of rhodamine-actin-injected wild-type and MZ*camsap2a* mutant embryos at 60% epiboly were analyzed by particle image velocimetry (PIV) (Fig. 4A).

Rose plots made from the PIV data showed that most of the actin flow in wild-type embryos was oriented animally (upward), as expected (Fig. 4B). MZ*camsap2a*[uot19] mutant embryos showed reduced upward and increased downward (vegetal) flow compared to wild type, while the reduction in upward flow was even more prominent in MZ*camsap2a*[uot20] embryos (Fig. 4B). Quantification of the fraction of vectors in the up, down, left and right directions revealed a significant difference in both mutant alleles compared to wild type (Fig. 4C,D). The PIV analysis also revealed that actin flow velocity was significantly reduced in MZ*camsap2a*[uot20] mutant embryos but not in MZ*camsap2a*[uot19] mutant embryos, consistent with MZ*camsap2a*[uot19] mutant embryos displaying a weaker phenotype (Fig. 4E). These results support the conclusion that the reduced and delayed actin accumulation in mutant embryos resulted from misoriented actin flow as well as, in the case of MZ*camsap2a*[uot20] mutants, slower flow.

We also investigated whether actin defects in mutant embryos were related to changes in actin dynamics by performing fluorescence recovery after photobleaching (FRAP) experiments. To examine yolk actin turnover, rhodamine-actin was injected into wild-type and MZ*camsap2a* mutant embryos and imaged by spinning disk confocal microscopy at mid-epiboly stages. Following photobleaching of a small region in the e-YSL close to the margin (Fig. 4F), there was no significant difference in the rate of fluorescence recovery in MZ*camsap2a* and wild-type embryos

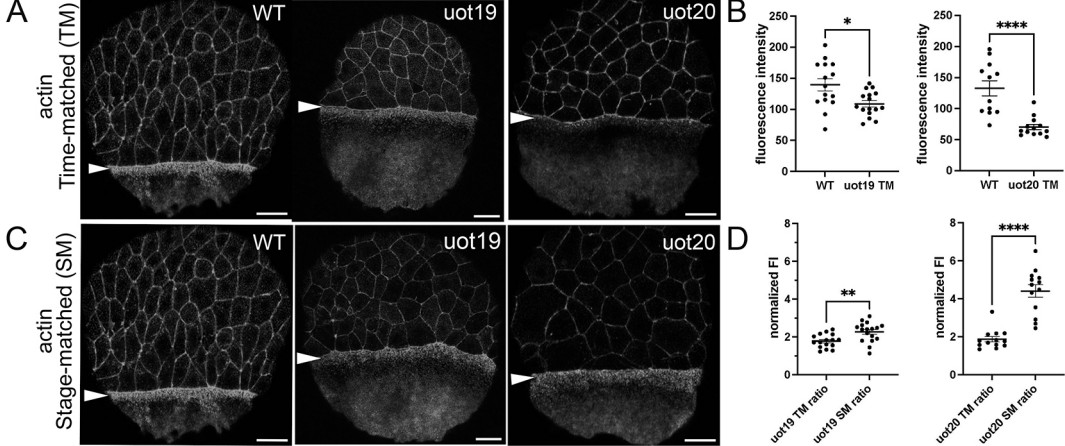

**Fig. 3. Actin accumulation in the e-YSL is reduced and delayed in mutant embryos.** (A,C) A phalloidin-stained wild-type embryo at 8 hpf (75% epiboly) shown alongside time-matched (A) and stage-matched (C) MZ*camsap2a*[uot19] and MZ*camsap2a*[uot20] embryos. Arrowheads indicate e-YSL actin band. (B,D) Quantification of e-YSL actin fluorescence intensity in wild-type (*n*=15), time-matched (uot19, *n*=16; uot20, *n*=13), (B) and stage-matched (uot19, *n*=17; uot20, *n*=13) mutant embryos (D). See Materials and Methods and Fig. S1C for details. Data are mean±s.e.m. *P=0.0192, **P=0.0032, ****P<0.0001 (two-tailed unpaired Mann-Whitney test). FI, fluorescence intensity; ST, stage matched; TM, time matched; WT, wild type. Scale bars: 50 μm.

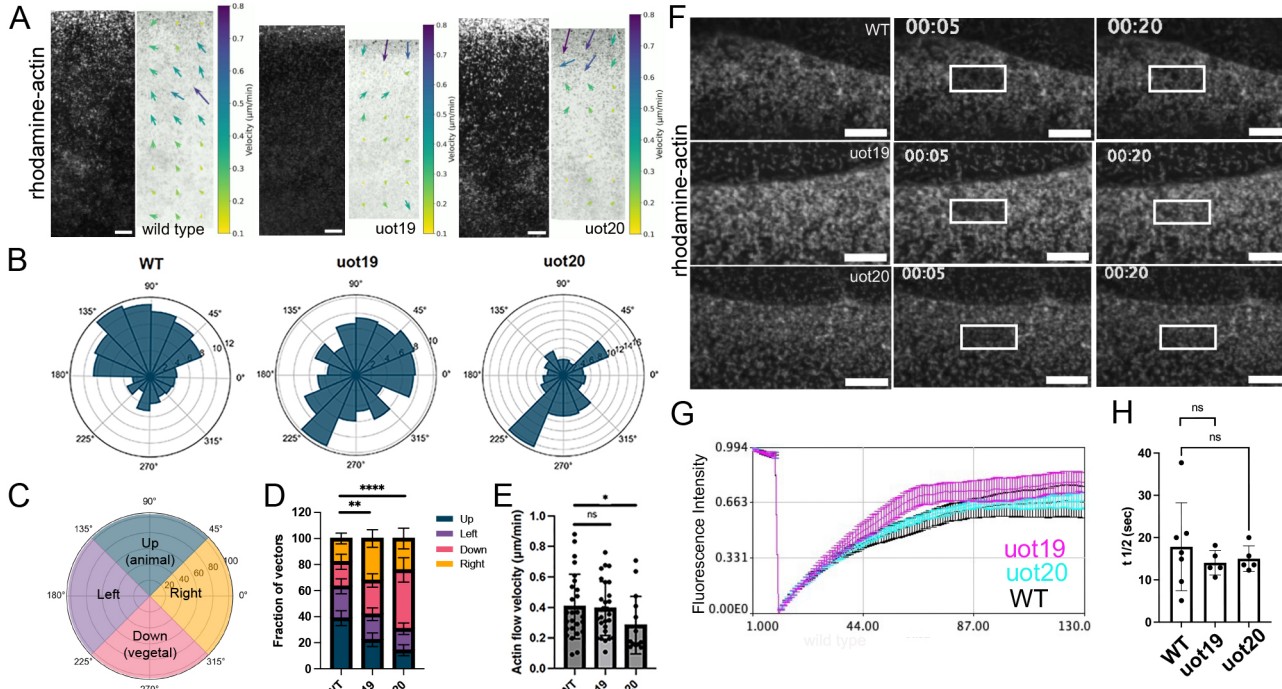

**Fig. 4. Yolk actin flow is misoriented in mutant embryos.** (A) Stills from spinning disk confocal time-lapse movies of rhodamine-actin in wild-type and MZ*camsap2a*<sup>uot19</sup> (uot19) and MZ*camsap2a*<sup>uot20</sup> (uot20) embryos at mid-epiboly stages. Actin flow quantified by PIV analysis, with arrows indicating direction and magnitude of flow. (B) Rose plots comparing rhodamine actin flow in wild-type and mutant embryos (wild type, *n*=34; uot19, *n*=21; uot20, *n*=17 embryos). (C) Quadrants used for analysis of vectors in D. (D) Distribution of PIV-derived actin flow vectors across the four quadrants in wild-type, uot19 and uot20 embryos. Error bars indicate s.e.m. **$P<0.0021$, ***$P<0.0001$ (one-tailed $\chi^2$ test). (E) Bar plot showing the actin flow velocity extracted from PIV analyses. *$P<0.05$ (two-sided Mann–Whitney test). (F) Stills from rhodamine-actin FRAP. Boxes indicate the photobleached region. (G) Fluorescence recovery curve, for the indicated genotypes. FRAP was performed at late epiboly stage in the e-YSL. (H) Quantification of half-time fluorescence recovery after FRAP. Data are mean±s.e.m. (Welch's two-tailed *t*-test). ns, not significant; WT, wild type. Scale bars: 10 μm.

(Fig. 4G,H), suggesting that the regulation of actin turnover in the e-YSL was unaffected in mutant embryos. Thus, our results suggest that defects in actin accumulation in the e-YSL in mutant embryos were primarily due to misoriented flow from the vegetal pole.

## Yolk myosin accumulation and actomyosin contractility are reduced in MZ*camsap2a* mutant embryos

Cortical actin and myosin flows have been shown to be coupled during epiboly (Behrndt et al., 2012); therefore, we hypothesized that myosin accumulation, by upward flow from the vegetal pole, was likely affected in mutant embryos. To investigate this, transgenic zebrafish embryos expressing GFP-tagged myosin [*Tg(actb2:myl12.1-GFP)*] (Maître et al., 2012) were crossed to MZ*camsap2a*<sup>uot20</sup> mutants to generate myosin-GFP transgenic MZ*camsap2a*<sup>uot20</sup> fish [*Tg(act2: myl12.1-GFP);*MZ*camsap2a*<sup>uot20/uot20</sup>]. Live control and time-matched *Tg(actb2:myl12.1-GFP);*MZ*camsap2a*<sup>uot20/uot20</sup> embryos were imaged by confocal microscopy (Fig. 5A,B). Changes in e-YSL myosin-GFP fluorescence intensity over time from mid to late epiboly stages were measured as a proxy for myosin accumulation during epiboly, due to potential differences in transgene expression levels (see Materials and Methods for details). We found that the rate of myosin accumulation in homozygous mutant transgenic embryos was significantly slower than that of controls (Fig. 5D). To support the conclusion that the reduced rate of myosin-GFP accumulation in MZ*camsap2a* mutants resulted from the loss of Camsap2a, myosin accumulation was also assessed in embryos heterozygous for *camsap2a*<sup>uot20</sup> [*Tg(actb2:myl12.1-GFP);*MZ*camsap2a*<sup>uot20/+</sup>] (Fig. 5C). The rate of myosin accumulation in *Tg(actb2:myl12.1-GFP);*MZ*camsap2a*<sup>uot20/+</sup> embryos was not significantly different

from that of control embryos (Fig. 5E), indicating that the reduction in marginal myosin level was associated with the Camsap2a homozygous mutation.

Actomyosin at the margin provides contractile force for epiboly (Behrndt et al., 2012); therefore, we next tested whether the reduction in marginal actin and myosin in mutant embryos resulted in lower mechanical tension, by using laser cutting. Horizontal UV laser cuts were used to assess tension produced by animally directed actin flow reflecting the activity of the flow-friction motor (Behrndt et al., 2012). Horizontal (perpendicular to the animal-vegetal axis) UV laser cuts were performed on rhodamine-actin injected wild-type embryos at 60% epiboly and time-matched mutant embryos (Fig. 6A, boxed regions). Tension in the e-YSL was measured by the initial recoil velocity along the animal-vegetal axis (Fig. 6A, arrows), which was calculated by the width of opening upon ablation over acquisition time (Fig. 6B). Our results showed that the rate of opening at horizontal cut sites was significantly reduced in both MZ*camsap2a* mutants (Fig. 6B). The reduction in horizontal tension in mutants is consistent with the abnormal animally directed actomyosin flow we observed. Overall, these results demonstrate that the accumulation of actin and myosin, facilitated by Camsap2a, is integral to the regulation and function of the actomyosin ring.

## Yolk cell macropinocytosis is reduced in MZ*camsap2a* mutant embryos

During epiboly, actin is important for the removal of yolk membrane by macropinocytosis, which is proposed to help maintain tension to balance mechanical forces between the blastoderm and yolk cell during epiboly (Cheng et al., 2023; Hernández-Vega et al., 2017;

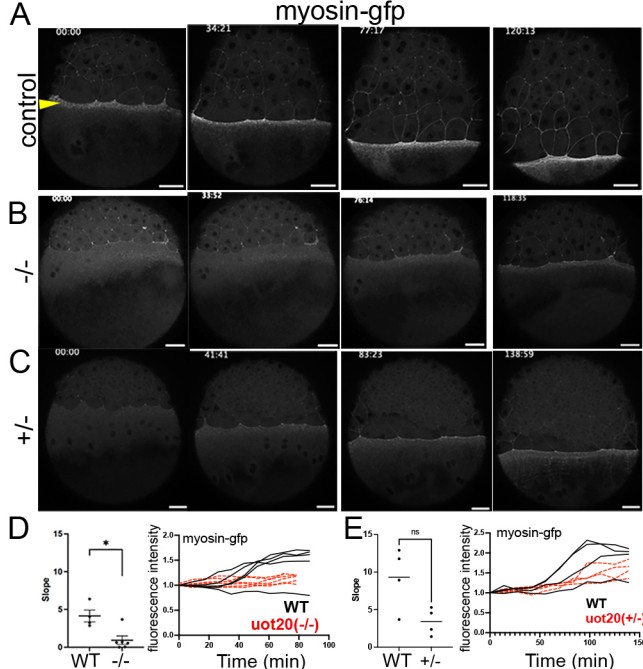

**Fig. 5. Yolk myosin-GFP accumulation is slower in mutant embryos.**
(A-C) Stills from time-lapse confocal movies during epiboly progression of
*Tg(actb2:myl12.1-GFP)* control (A), *Tg(actb2:myl12.1-GFP);*
MZ*camsap2a^uot20/uot20* (B) and *Tg(actb2:myl12.1-GFP);Mcamsap2a^uot20/+*
(C) embryos. Time stamp indicates min:s. Arrowhead indicates e-YSL. Scale
bars: 50 μm. (D) Quantification of myosin accumulation in the e-YSL over
time in *Tg(actb2:myl12.1-GFP);*MZ*camsap2a^uot20/uot20* compared to
*Tg(actb2:myl12.1-GFP)* (uot20−/−) embryos. Left panel shows the slope of
the lines shown in the right panel. (E) Quantification of e-YSL myosin
accumulation over time in *Tg(actb2:myl12.1-GFP);*MZ*camsap2a^uot20/+*
(uot20+/−) compared to *Tg(actb2:myl12.1-GFP)* embryos, as in D. Data are
mean±s.e.m. Welch's *t*-test. *P=0.0163, ^nsP=0.0563. ns, not significant; WT,
wild type.

Trinkaus, 1984). Macropinocytosis is confined to the e-YSL where
the actin ring forms and where extensive actin-dependent membrane
ruffles are also localized (Cheng et al., 2023; Solnica-Krezel and
Driever, 1994). Since we observed actin defects in the e-YSL in
mutant embryos, we examined whether macropinocytosis and
membrane ruffling were also disrupted.

We performed a detailed time-course analysis of macropinocytosis,
which is linked to membrane ruffling (Cheng et al., 2023). Wild-
type and time-matched mutant embryos at 4.7-5 hpf (30-40%
epiboly), 6 hpf (shield) (Fig. 7A,D) and 8.5-9 hpf (75-80% epiboly)
(Fig. 7B,E) stages were bathed in FITC-dextran, which is taken up
into the e-YSL by macropinocytosis (Marsal et al., 2021; Solnica-
Krezel and Driever, 1994). During mid-epiboly stages, the number

of internalized fluorescent vesicles was variable in wild-type and
mutant embryos (Fig. 7C,F). After 7 hpf, a significant reduction in
the number of internalized vesicles was observed in mutant
embryos, corresponding to the time when the epiboly delay was
prominent (Fig. 7C,F). These observations suggested that the yolk
macropinocytosis defects in mutant embryos worsened as epiboly
progressed. Since MZ*camsap2a* mutant embryos were able to
recover and progress to late epiboly stages, mutant embryos that
were stage-matched to 75% epiboly wild-type embryos were also
assessed to check whether macropinocytosis recovered at later
stages. Our results showed that the number of endocytosed vesicles
in stage-matched mutant embryos for both alleles did not recover
over the period examined but remained significantly lower than
wild-type embryos (Fig. 7C,F, compare wild type at 8.5 hpf to
mutants at 9 and 9.5 hpf).

We also examined membrane ruffling in wild-type and
MZ*camsap2a* mutant embryos from both alleles that were time-
and stage-matched to 75% epiboly stage wild-type embryos by
scanning electron microscopy. This analysis revealed abnormal
e-YSL membrane ruffles that appeared to be shallower compared to
wild-type embryos (Fig. S1E). In both time- and stage-matched
embryos, the membrane ruffles looked different in mutants
compared to wild-type embryos, suggesting that the defect does
not improve over time. These results suggest an essential role for
Camsap2a in yolk cell macropinocytosis, although the reduced
levels of macropinocytosis did not prevent the delayed completion
of epiboly.

## Yolk-specific expression of Camsap2a rescues epiboly delay, actin accumulation and yolk macropinocytosis in mutant embryos

To confirm that the phenotypes we observed were due to the loss
of Camsap2a, we performed rescue experiments with full-length
Camsap2a. Given that the *camsap2a* transcript is confined to the
YSL during epiboly, we examined whether yolk cell-specific
expression of Camsap2a could rescue the MZ*camsap2a^uot20*
mutant phenotype. We focused on MZ*camsap2a^uot20* because this
allele produced embryos with more severe phenotypes than the
MZ*camsap2a^uot19* allele. To accomplish this, a plasmid containing
the *wnt8* YSL-specific promoter (Narayanan and Lekven, 2012)
driving the expression of Camsap2a was injected into the yolk
cell of mutant and wild-type embryos at the one-cell stage. To
determine the rescue ability of the construct, we examined epiboly
progression, actin accumulation and macropinocytosis in the yolk
cell.

Epiboly progression was assessed by performing whole-mount
*in situ* hybridization for the marginal mesodermal marker *T-box
transcription factor Ta* (*tbxta*), as previously described (Schulte-
Merker et al., 1994; West et al., 2017) (see Materials and Methods

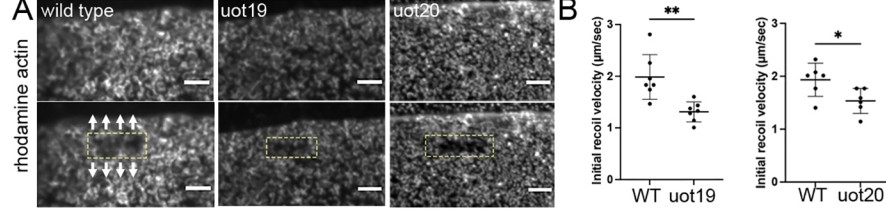

**Fig. 6. Tension is reduced in the actomyosin ring in mutant embryos.** (A) Stills from horizontal UV laser cuts in the e-YSL of wild-type and mutant
embryos injected with rhodamine-actin. Top row: pre-cut; bottom row: post-cut (marked by boxed region). Arrows indicate direction of recoil along the animal-
vegetal axis. Scale bars: 10 μm. (B) Initial recoil velocities for UV laser cuts perpendicular (horizontal) to the animal-vegetal axis in wild-type (*n*=6),
MZ*camsap2a^uot20* (uot20; *n*=5) and MZ*camsap2a^uot19* (uot19; *n*=7) embryos. Data are mean±s.e.m. **P*=0.03636, ***P*=0.0052 (Welch's *t*-test).

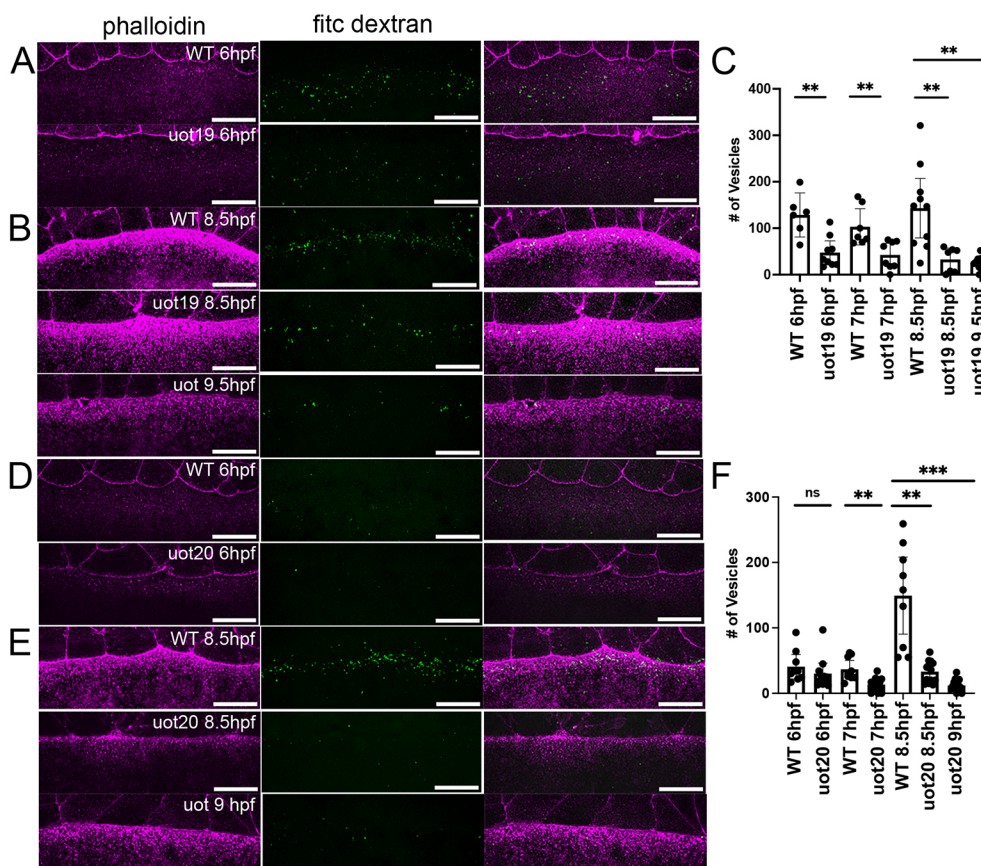

**Fig. 7. Macropinocytosis is reduced in mutant embryos.** (A,B,D,E) Confocal images of phalloidin (magenta) and FITC-dextran puncta (green) in wild-type and mutant embryos at the times indicated. Scale bars: 10 μm. (C) Quantification of FITC-dextran puncta the e-YSL in wild-type and MZ*camsap2a^{uot19}* mutant embryos at the indicated time points. (WT 6/7/8.5 hpf, *n*=8,7,11; uot19 6 hpf, *n*=9; uot19 7 hpf, *n*=8; uot19 8.5 hpf, *n*=8; uot19 9.5 hpf, *n*=12). Data are mean±95%CI. **P<0.05 (Welch's *t*-test). (F) Quantification of FITC-dextran puncta in the e-YSL in wild-type and MZ*camsap2a^{uot20}* mutant embryos at the indicated time points. (WT 6/7/8.5 hpf, *n*=9; uot20 6 hpf, *n*=11; uot20 7 hpf, *n*=16; uot20 8.5 hpf, *n*=13; uot20 9 hpf, *n*=15). Data are mean±s.e.m. **P=0.0038 (7 hpf versus uot20 7 hpf), **P=0.0017 (8.5 hpf versus uot20 8.5 hpf), ***P=0.0006 (Welch's *t*-test). ns, not significant; WT, wild type.

and Fig. S1F). Yolk-specific expression of Camsap2a had no effect on epiboly progression in wild-type embryos but resulted in a significant rescue of the epiboly delay in mutant embryos (Fig. 8A,A′). Similarly, Camsap2a overexpression did not alter actin accumulation in the e-YSL of wild-type embryos (Fig. 8B,B′). In MZ*camsap2a^{uot20}*-injected embryos, actin levels were restored to wild-type levels (Fig. 8B,B′). We also noticed that EVL actin levels increased in the rescued embryos, consistent with the reduction in EVL actin being secondary to yolk defects (Fig. 8B).

We next examined whether the actin rescue was accompanied by restoration of actin-dependent membrane internalization. In wild-type embryos, yolk-specific Camsap2a expression did not significantly alter yolk macropinocytosis (Fig. 8C,C′). However, in MZ*camsap2a^{uot20}* mutants, yolk-specific expression of Camsap2a partially restored macropinocytosis, as shown by the significant increase in internalized fluorescent dextran particles compared to uninjected mutants (Fig. 8C′). These data indicate that the defects observed in MZ*camsap2a^{uot20}* mutants are a consequence of reduced Camsap2a in the yolk cell.

### Constitutively active Rab5ab partially rescues the MZ*camsap2a* mutant phenotype

We next set out to identify the molecular pathway by which Camsap2a mediates membrane dynamics via the actin cytoskeleton. The small GTPase Rab5 is required for macropinocytosis and can also regulate actin dynamics via downstream effectors (Lanzetti et al., 2004). In zebrafish, morpholino studies have shown that yolk-specific knockdown of *rab5ab* causes elongated embryo morphology, epiboly delay and reduced macropinocytosis in the e-YSL (Kenyon et al., 2015; Marsal et al., 2021). In addition, yolk actin flow is slower and misoriented in Rab5ab morphants

leading to reduced actomyosin accumulation (Marsal et al., 2021). Strikingly, all the defects seen in *rab5ab* morphant embryos are also observed in *camsap2a* mutant embryos, suggesting a potential mechanistic link and, further, that a primary defect in macropinocytosis can lead to actomyosin flow defects and epiboly delay. In addition, the *C. elegans* Camsap homolog PTRN-1 partially colocalizes with Rab5-positive endosomes in the intestine (Gong et al., 2018). Human CAMSAP2 can physically interact with several Rab-GEFs and has been proposed to activate them (Boldt et al., 2016; Go et al., 2021; Ho et al., 2023). Taken together, these findings led us to hypothesize that Camsap2a might regulate Rab5ab in the yolk cell during epiboly by facilitating its activation during macropinosome formation.

If Camsap2a functions upstream of Rab5ab to regulate macropinocytosis, then exogenous expression of Rab5ab might lead to increased levels of active Rab5ab and rescue the phenotype. For this analysis, we used MZ*camsap2a^{uot20}* mutant embryos, and we first examined the ability of Rab5ab to rescue the macropinocytosis defects when Rab5ab was exogenously expressed specifically in the yolk cell. Rab5ab-injected embryos were soaked in FITC-dextran as described above and we found that expression of Rab5ab led to an increase in FITC-dextran-positive vesicles in wild-type embryos but had no effect on mutant embryos (Fig. 9A,A′). We then reasoned that if Rab5ab activation depends primarily upon Camsap2a, it might be necessary to use a constitutively active form of Rab5ab (CA-Rab5ab) for rescue. We found that the number of FITC-dextran positive vesicles was increased in both wild-type and mutant embryos expressing CA-Rab5ab (Fig. 9B,B′). These findings support our hypothesis that Camsap2a is required for Rab5ab activity during macropinocytosis in the yolk cell.

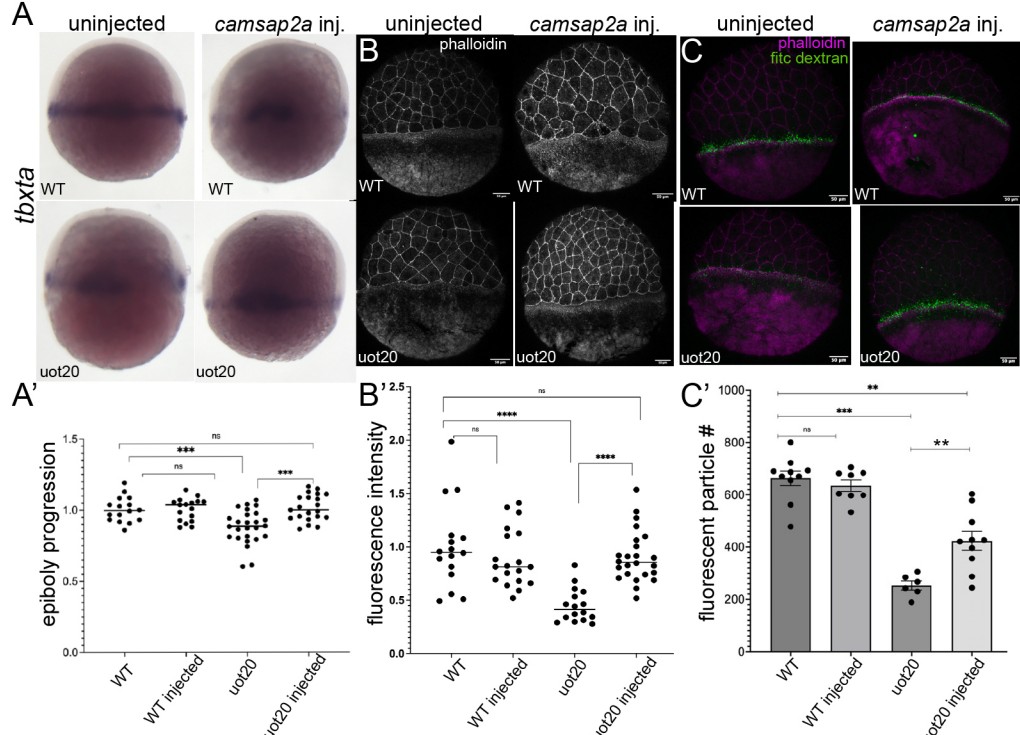

**Fig. 8. Yolk expression of Camsap2a rescues mutant phenotypes.** (A) *In situ* hybridization for *tbxta* in uninjected and *camsap2a*-injected wild-type and MZ*camsap2a^uot20^* mutant embryos at 7 hpf. (A′) Quantification of epiboly progression (see Fig. S1F and Materials and Methods). Data are presented as dot plots of individual embryos; horizontal lines indicate the median. Wild type (*n*=16); wild-type injected (*n*=17); MZ*camsap2a^uot20^* (*n*=26); MZ*camsap2a^uot20^* injected (*n*=21). Two trials, each normalized to the respective wild-type average. ***P*=0.0008 (two-tailed Mann–Whitney test) (*N*=2). (B) Phalloidin-stained uninjected and *camsap2a*-injected wild-type and MZ*camsap2a^uot20^* mutant embryos at 7 hpf. (B′) Quantification of e-YSL phalloidin fluorescence intensity. Data are presented as dot plots of individual embryos; horizontal lines indicate the median. Wild type (*n*=16); wild-type injected (*n*=19); MZ*camsap2a^uot20^* (*n*=16); MZ*camsap2a^uot20^* injected (*n*=24). Three trials, each normalized to respective wild-type average. *****P*<0.0001 (two-tailed Mann–Whitney test) (*N*=3). (C) Phalloidin (magenta) and FITC-dextran (green) in uninjected and injected wild-type and mutant MZ*camsap2a^uot20^* embryos. (C′) Quantification of FITC-dextran particles in the e-YSL in uninjected and injected wild-type and MZ*camsap2a^uot20^* embryos. Wild type (*n*=10); wild-type injected (*n*=8); MZ*camsap2a^uot20^* (*n*=6); MZ*camsap2a^uot20^* injected (*n*=10). Data are presented as mean±s.e.m. ***P*<0.003, ****P*=0.0001, ^ns^*P*=0.4 (two-tailed Mann–Whitney test). ns, not significant. uot20, MZ*camsap2a^uot2^*; WT, wild type. Scale bars: 50 μm.

We then investigated whether actin accumulation in mutant embryos could be rescued by Rab5ab. Phalloidin staining of wild-type embryos at 7 hpf (60% epiboly) and time-matched mutant embryos showed that, in both cases, yolk-specific overexpression of Rab5ab did not alter actin accumulation in the e-YSL (Fig. 9C,C′). Expression of CA-Rab5ab in wild-type embryos resulted in reduced actin accumulation compared to uninjected wild-type controls. In contrast, CA-Rab5ab restored actin levels in mutant embryos, although not to wild-type levels (Fig. 9D,D′).

We then examined whether the overall epiboly delay in mutant embryos was rescued by Rab5ab. To test this possibility, epiboly progression in wild-type and MZ*camsap2a^uot20^* mutant embryos exogenously expressing Rab5ab and CA-Rab5ab was determined by *in situ* hybridization for *tbxta* (Schulte-Merker et al., 1994; West et al., 2017). Our results showed that Rab5ab overexpression had no effect on epiboly in wild-type and mutant embryos (Fig. 9E,E′). CA-Rab5ab-expressing wild-type embryos were mildly delayed during epiboly progression while epiboly was rescued in CA-Rab5ab expressing MZ*camsap2a^uot20^* mutant embryos (Fig. 9F,F′). Thus, CA-Rab5ab was able to rescue macropinocytosis, actin accumulation and epiboly in MZ*camsap2a* mutant embryos, consistent with our hypothesis that Camsap2a functions upstream of Rab5ab in the yolk cell during epiboly.

## DISCUSSION

Epiboly is a widely conserved cell movement during animal development in which a cell sheet or multilayered cell mass thins and spreads. In zebrafish, the embryonic blastoderm and YSL undergo epiboly to enclose the extra-embryonic yolk cell by the end of gastrulation. Work from many groups has established the central role of the e-YSL actomyosin contractile ring as the driver of zebrafish epiboly (Bruce and Heisenberg, 2020). Actin and myosin flow upwards from the vegetal pole and assemble into a contractile band by mid-epiboly stages (Behrndt et al., 2012). The attachment of the outer epithelial layer of the blastoderm to the e-YSL allows it to be pulled vegetally by the contracting ring. In addition, a region of macropinocytosis in the e-YSL, which removes excess yolk cell membrane as epiboly proceeds, is thought to contribute to balancing the forces between the blastoderm and yolk cell – a balance needed to drive timely epiboly (Cheng et al., 2023; Marsal et al., 2021). What remains unclear is how the events in the yolk cell are regulated. We identified Camsap2a, a member of a protein family known to bind non-centrosomal microtubule minus ends, as an important regulator of epiboly via its effects on yolk cell macropinocytosis and actomyosin contractile ring formation.

Epiboly progression and formation of the actomyosin ring in the e-YSL are delayed in MZ*camsap2a* mutant embryos, while yolk

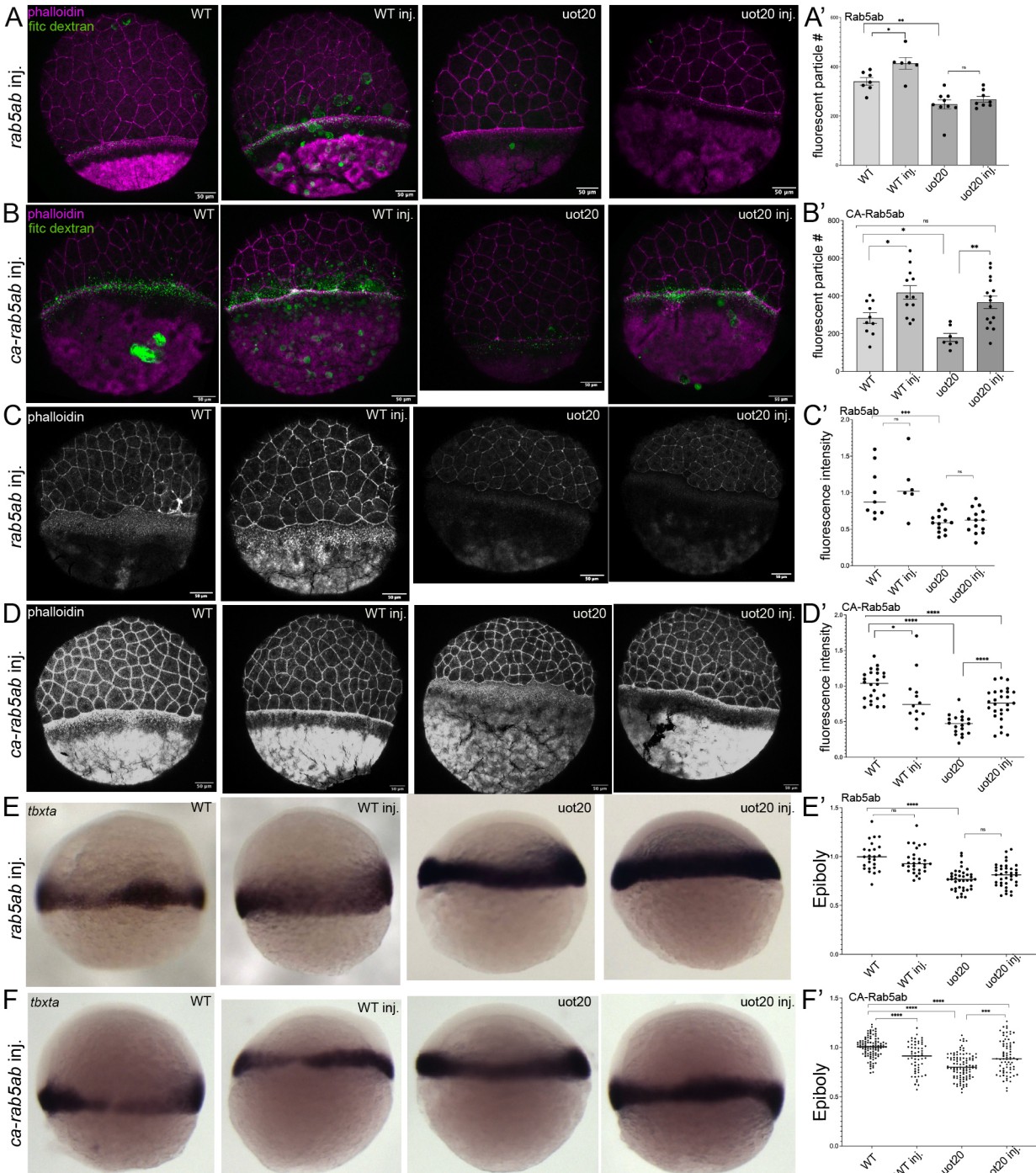

**Fig. 9.** See next page for legend.

cell microtubules appear largely normal. PIV analysis showed that actin flow is misoriented compared to wild-type embryos and laser-cutting experiments showed that the ring is under less tension. We also found that macropinocytosis in the e-YSL is reduced in mutant embryos. Additionally, we showed that the mutant phenotypes could be rescued by yolk-specific expression of full-length Camsap2a. Overall, these results point to a crucial role for Camsap2a in the yolk cell during epiboly.

To understand how Camsap2a functions in the yolk cell, we noted the striking overlap between the MZ*camsap2a* mutant phenotype, and the phenotype observed after yolk-specific morpholino

knockdown of *rab5ab* (Kenyon et al., 2015; Marsal et al., 2021). *rab5ab* morphants exhibit impaired epiboly, reduced actomyosin flow, accumulation, and contractility as well as reduced macropinocytosis. In addition, *rab5ab* morphant embryos are often elongated along the animal-vegetal axis, as also seen in MZ*camsap2a* mutant embryos and indicative of altered tensions within the embryo (Marsal et al., 2021). Rab5 plays a conserved role in macropinocytosis, suggesting that reducing macropinocytosis in the yolk cell could lead to the actomyosin defects observed in morphants. We hypothesized that Camsap2a regulates Rab5ab activity, which was supported by the rescue of the epiboly delay,

**Fig. 9. CA-Rab5ab rescues macropinocytosis, actin accumulation and epiboly in mutant embryos.** (A) Phalloidin (magenta) and FITC-dextran (green) in uninjected and *rab5ab*-injected wild-type and MZ*camsap2a^uot20* embryos. (A′) Quantification of FITC-dextran in e-YSL in wild type (*n*=7), *rab5ab*-injected wild-type (*n*=6), MZ*camsap2a^uot20* (*n*=9) and MZ*camsap2a^uot20* embryos (*n*=8). Data are presented as mean±s.e.m. *P*=0.02, **P*=0.003, ^ns*P*=0.4 (two-tailed Mann–Whitney test) (*N*=1). (B) Phalloidin (magenta) and FITC-dextran (green) in uninjected and *ca-rab5ab*-injected wild-type and MZ*camsap2a^uot20* embryos. (B′) Quantification of FITC-dextran particles in E-YSL in wild type (*n*=10), *ca-rab5ab*-injected wild type (*n*=12), MZ*camsap2a^uot20* (*n*=7) and *ca-rab5ab*-injected MZ*camsap2a^uot20* embryos (*n*=15). Data are presented as mean±s.e.m. *P*=0.01 (wt versus wt injected), *P*=0.02 (wt versus uot20), **P*=0.0015, ^ns*P*=0.08 (two-tailed Mann–Whitney test) (*N*=1). (C) Phalloidin staining in uninjected and *rab5ab*-injected wild-type and MZ*camsap2a^uot20* mutant embryos at 7 hpf. (C′) Quantification of phalloidin fluorescence intensity in e-YSL in wild type and mutant uninjected and *rab5ab*-injected embryos (two trials, each normalized to the respective wild-type average). Data are presented as dot plots of individual embryos; horizontal lines indicate the median. Wild type (*n*=9); wild-type injected (*n*=6); MZ*camsap2a^uot20* (*n*=14); MZ*camsap2a^uot20* injected (*n*=14). ***P*<0.0009 (two-tailed Mann–Whitney test) (*N*=2). (D) Phalloidin staining in uninjected and *ca-rab5ab*-injected wild-type and MZ*camsap2a^uot20* mutant embryos at 7 hpf. (D′) Quantification of phalloidin fluorescence intensity in e-YSL in wild-type and mutant uninjected and *rab5ab*-injected embryos (three trials, each normalized to the respective wild-type average). Data presented as dot plots of individual embryos, horizontal lines indicate the median. Wild type (*n*=24); wild-type injected (*n*=12); MZ*camsap2a^uot20* (*n*=19); MZ*camsap2a^uot20* injected (*n*=27). *P*=0.03, ****P*<0.0001 (two-tailed Mann–Whitney test) (*N*=3). (E) *In situ* hybridization for *tbxta* in uninjected and *rab5ab*-injected wild-type and MZ*camsap2a^uot20* mutant embryos. (E′) Quantification of epiboly progression in uninjected and *rab5ab*-injected embryos as in Fig. 8A. Data are presented as dot plots of individual embryos; horizontal lines indicate the median. Wild type (*n*=25); wild-type injected (*n*=29); MZ*camsap2a^uot20* (*n*=38); MZ*camsap2a^uot20* injected (*n*=37). ****P*<0.0001, ^ns*P*=0.1612 (wild type to wild-type injected) (*N*=1), ^ns*P*=0.076 (mutant to mutant injected) (*N*=2) (two-tailed Mann–Whitney test). (F) *In situ* hybridization for *tbxta* in uninjected and *ca-rab5ab*-injected wild-type and MZ*camsap2a^uot20* mutant embryos. (F′) Quantification of epiboly progression in uninjected and injected embryos as in Fig. 8A (three trials, each normalized to respective wild type average). Data are presented as dot plots of individual embryos; horizontal lines indicate the median. Wild type (*n*=100); wild-type injected (*n*=57); MZ*camsap2a^uot20* (*n*=103); MZ*camsap2a^uot20* injected (*n*=73). ***P*<0.001, ****P*<0.0001 (two-tailed Mann–Whitney test) (*N*=5). inj., injected; ns, not significant; uot20, MZ*camsap2a^uot2*; WT, wild type. Scale bars: 50 µm.

macropinocytosis, and actin accumulation defects following yolk-specific expression of constitutively active Rab5ab in MZ*camsap2a^uot20* mutant embryos.

Tension within the e-YSL increases over the course of epiboly, resulting in a tension gradient along the yolk animal-vegetal axis, which is required for cortical flow of actin and myosin to the e-YSL (Behrndt et al., 2012). It has been proposed that Rab5ab-mediated membrane removal is required to achieve the necessary level of yolk cortical tension to regulate epiboly progression (Marsal et al., 2021), explaining how a disruption in macropinocytosis could lead to defects in actomyosin accumulation. Although additional work is required to understand how Campsa2a interacts with Rab5ab, our model is that Camsap2a functions to regulate Rab5ab activity, which it may do by activating the GEF for Rab5ab. Given the known role of Rab5 during late stages of macropinosome scission (Maxson et al., 2021; Salloum et al., 2023), the loss of Camsap2a might result in stalled macropinosome formation. In this scenario, the dismantling of actin during scission might be impaired in mutant embryos, leading to overall reduced availability of actin monomers for the actomyosin contractile motors. Future work is required to determine the localization patterns and dynamics of both Camsap2a

and Rab5 proteins in the zebrafish yolk cell to further understand how they might interact.

Not all aspects of the *camsap2a* phenotype were fully rescued by CA-Rab5ab. For example, actin accumulation was not rescued to wild-type levels, suggesting that Camsap2a may also have Rab5ab-independent functions in the yolk cell. We have also not ruled out a role for microtubules in the Camsap2a mutant phenotype. Microtubule binding of Camsap2a might play a role in localizing factors such as Rab-GEFs. However, recently Camsaps in several contexts have been shown to have microtubule-independent functions, particularly in trafficking. For example, in the *C. elegans* intestinal epithelium, the Camsap homolog PTRN-1 functions independently of microtubules to activate formin, which nucleates unbranched actin polymerization involved in recycling clathrin-independent cargo (Gong et al., 2018). Interestingly, the CH domain of PTRN-1 was shown bind to formin (Gong et al., 2018). Although future experiments are required, we postulate that the CH domain of Camsap2a might also be essential for actin regulation in zebrafish since the *camsap2a^uot19* allele is a 12 bp in-frame deletion in the CH domain and produces a similar phenotype as the *camsap2a^uot20* allele, which encodes a stop codon in the CH domain. However, the *camsap2a^uot19* is milder than the *camsap2a^uot20* phenotype, suggesting that other regions of the protein are also important. In preliminary work, we found that expression of the CH domain of Camsap2a can partially rescue actin accumulation in mutant embryos (S.Q., unpublished data). As few studies have examined the function of the CH domain in Camsap proteins, this will be an important direction for future work.

Our current model is that Camsap2a functions in the e-YSL to activate Rab5ab, which is required for yolk membrane macropinocytosis during epiboly progression. It is proposed that macropinocytosis contributes, along with the actomyosin motors in the e-YSL, to modulation of yolk cell cortical tension during epiboly. An important open question is the upstream regulation of Camsap2a, which we hypothesize may be via calcium signaling. Calcium waves in the e-YSL have been linked to activation of actomyosin contractility and more recently to macropinocytosis (Cheng et al., 2004, 2023). The CC1 domain of Camsap proteins is the second most conserved domain and the mammalian Camsap1 CC1 domain can bind calmodulin (King et al., 2014). In addition, the mammalian Camsap2 gene regulatory region contains a calcium-response element that is transcriptionally activated in the nervous system by the calcium-response factor (CaRF) transcription factor (West, 2011). One possibility to explore in the future is that calcium signaling might regulate zygotic expression of Camsap2a in the YSL.

Our work has provided new insights into the role of a Camsap family member during vertebrate development as well as the molecular control of epiboly. These findings further implicate regulation of membrane dynamics as an important contributor to the process. Membrane dynamics are also linked to other morphogenetic processes (Clark et al., 2014), such as apical constriction of bottle cells (Lee and Harland, 2007) and endoderm morphogenesis in the *Xenopus* gastrula (Wen and Winklbauer, 2017), as well as during neural crest migration in the chick embryo (Li et al., 2020). Our study adds to a growing body of work pointing to the widespread importance of membrane dynamics during animal morphogenesis, as well as revealing new developmental functions of a member of the Camsap family.

## MATERIALS AND METHODS
### Zebrafish handling
Animals were maintained in accordance with the policies and procedures of the University of Toronto animal care committee. Fish were housed at

28-30°C in an Aquaneering Zebrafish Housing System with the pH from 7.2 to 7.8 and conductivity between 600 and 700 µS. Adults were on average 1 year old with no prior manipulations or health issues. Lines used were: AB wild type, *Tg(actb2:myl12.1-GFP)* (Maître et al., 2012), MZ*camsap2a^uot19^*, MZ*camsap2a^uot20^*, *Tg(actb2:myl12.1-GFP)*;MZ*camsap2a^uot20/uot20^* and *Tg(actb2:myl12.1-GFP)*;MZ*camsap2a^uot20/+^*. Embryos were acquired from natural spawnings, kept in fresh fish facility water and staged as described by Kimmel et al. (1995).

## CRISPR/Cas9 mutant generation

To generate *camsap2a* CRISPR knockout mutants, the desired single-stranded guide RNA (sgRNA) sequence (T7 promoter: TTCTAATA-CGACTCACTATA; target sequence: GTCAGCAGGTTGTCCACCGGA; overlap sequence: GTTTTAGAGCTAGA) was designed using CRISPR-scan (https://www.crisprscan.org) (Gagnon et al., 2014) and targets exon 5, which contains the CH domain. The sgRNA template was generated by PCR as described (Gagnon et al., 2014) (Oligo 1: 5′-TTCTAATACGACTCAC-TATAGTCAGCAGGTTGTCCACCGGAGTTTTAGAGCTAGA-3′; Oligo 2: 5′-AAAAGCACCGACTCGGTGCCACTTTTTCAAGTTGATAACG-GACTAGCCTTATTTTAACTTGCTATTTCTAGCTCTAAAAC-3′).

The assembled guide RNA template was transcribed using the MegaScript T7 Transcription kit (Thermo Fisher Scientific, AM1334). *cas9* mRNA was synthesized by digesting pT3TS-nCas9n (Addgene plasmid #46757, deposited by Wenbiao Chen; RRID:Addgene_46757) (Jao et al., 2013) with XbaI (NEB, R0145) and transcribing with the mMESSAGE mMachine T3 Transcription Kit (Thermo Fisher Scientific, AM1348); 50 pg sgRNA and 300 pg *cas9* mRNA were co-injected into one-cell-stage embryos. Genotyping was carried by extracting genomic DNA from 24 hpf embryos using Lysis Buffer (10 mM Tris, pH 8, 10 mM EDTA, 210 mM NaCl, 0.5% SDS, 200 µg/ml proteinase K), phenol/chloroform extraction and ethanol precipitation. The targeted region was PCR amplified with Taq polymerase with ThermoPol buffer (NEB, M0267S) using the forward primer 5′-GGGGATCTTTGACTGTTGCC-3′ and the reverse primer 5′-TGTTAATCCAGGATCACAACGT-3′. Amplified PCR fragments were digested with BamHI-HF (NEB, R3136) to determine the genotype of the embryo. Samples showing heterozygous and homozygous genotypes were purified using PEG8000 (Promega, V3011) and sent for sequencing to validate the mutation. Sequencing was done by The Centre of Applied Genomics and Eurofins Genomics.

Two mutant alleles were generated using CRISPR/Cas9: *camsap2a^uot19^*, which contains a 12 bp in-frame deletion; and *camsap2a^uot20^*, which contains a 13 bp deletion (stop codon underlined). WT: ACCGTGGAT-CCCTCCGGTGGACAACCTGCTGAAGGACAGCACAGA; camsap2a^uot19^: ACCGTGGATC————————AACCTGCTGAAGGACAGCAC-AGA; camsap2a^uot20^: ACCGTGGA—————————CAACCT-GCTGAAGGACAGCACAAA.

## RT-PCR for *camsap2a*

To assess maternal deposition of *camsap2a* transcripts, RT-PCR was performed on cDNA made from cleavage (8- to 16-cell stage), 7 hpf and 8-9 hpf wild-type embryos using TRIzol Reagent (Thermo Fisher Scientific, 15596026) and the Protoscript First Strand cDNA Synthesis Kit (NEB, E6300S) following the manufacturer's instructions. PCR amplification was done with the Q5 High Fidelity 2X Master Mix (NEB, M0492S). *actb1* was used as a control and the primers were: forward primer, 5′-ATGGATGAGG-AAATCGCTGC-3′; reverse primer, 5′-CACAGCTTCTCCTTGATGTC-3′. The CKK domain of *camsap2a* was amplified using the following primers: forward primer, 5′-ACGCTCGAGATGGGTCCTAAATTATACAAAG-AG-3′ (contains XhoI restriction site); reverse primer, 5′-CCGTCT-AGAGGTCTAGGACTTGACAGCCGCTAC-3′ (contains XbaI restriction site).

## Quantitative real-time PCR (qPCR) analysis

RNA from three batches of shield-stage wild-type embryos and three batches of MZ*camsap2a^uot20^* embryos was isolated using TRIzol reagent following the manufacturer's instructions. Each batch contained 50 embryos. The RNA was treated with Monarch DNaseI (NEB, T2104) to remove genomic DNA and was purified using the Monarch Spin RNA

Isolation Kit (NEB, T2110S) following the manufacturer's instructions; 1 µg total RNA was used to make cDNA with the LunaScript SuperMix kit (NEB, E3010S) following the manufacturer's instructions.

qPCR was performed in three technical replicates using SsoAdvanced Universal SYBR Green Supermix (Bio-Rad, 1725271) in hard-shell, thin-wall, 384-well skirted PCR plates (Bio-Rad, HSP3805) sealed with ABsolute qPCR Seal (Thermo Fisher Scientific, AB-1170) in a CFX Opus Thermocycler (Bio-Rad, 1201452). Pooled wild-type and mutant cDNA samples were used to generate a standard curve. mRNA expression levels of Camsap orthologs (*camsap1a*, *camsap1b*, *camsap2a*, *camsap2b* and *camsap3*) relative to *lsm12b* expression were assessed by qPCR using following primers as described (Taylor et al., 2019; Willoughby et al., 2021): camsap1a_qFP, 5′-CTGTGATGGAGGATCTGATG-3′; camsa-p1a_qREP, 5′-CACACTGTCCGATAGAGAC-3′; camsap1b_qFP, 5′-GT-GACCTCAATAGCAGACAG-3′; camsap1b_qREP, 5′-GAGCTCCGCT-ATAAATACCAT-3′; camsap2a_qFP, 5′-GCCCATCTCTAAAGTCAC-AA-3′; camsap2a_qREP, 5′-GAAGTCTGTGGTTGAGGTC-3′; cam-sap2b_qFP, 5′-GACTCACTTAACAAGGCCAG-3′, camsap2b_qREP, 5′-TGGTCCATGTCTGAGTAGTC-3′; camsap3_qFP, 5′-GCCAAGCTTC-AAGTCCAATA-3′; camsap3_qREP, 5′-AGGATGAGGAAATGGTT-TGC-3′; lsm12b_qFP, 5′-AGTTGTCCCAAGCCTATGCAATCAG-3′; lsm12b_qREP, 5′-CCACTCAGGAGGATAAAGACGAGTC-3′.

## Whole-mount in situ hybridization

Whole-mount *in situ* hybridization was performed as described (Jowett and Lettice, 1994; West et al., 2017). A *tbxta* digoxigenin-labeled antisense riboprobe was made by T7 *in vitro* transcription of XhoI-digested plasmid, using the Dig RNA labeling kit (Millipore Sigma, 11175025910) following the manufacturer's instructions. The probe was purified using NucAway Spin Columns (Ambion, AM10070) or the Monarch RNA Clean Up Kit (NEB, T2030L), following the manufacturers' instructions.

For *campsap2a*, a fragment from 618-1724 bp of the coding region was PCR amplified (Q5 2x Master Mix, NEB, M0492L) from 1 dpf cDNA with the forward primer 5′-CGCAGAGCCTGTTGAGAATCC-3′ and reverse primer 5′-GATGAGTCCTCGTCGAGTGTC-3′ (Winata et al., 2018). The fragment was purified, A-tailed and ligated into pGEM-T Easy (Promega, A1360) following the manufacturer's instructions and confirmed by sequencing. The *camsap2a* digoxigenin-labeled antisense riboprobe was made by digesting the plasmid with NdeI (NEB, R0111S) and transcribing with T7 RNA polymerase using the Dig RNA labeling kit. The probe was purified using NucAway Spin Columns.

## Cloning, plasmids and constructs

RNAs for injection were synthesized from linearized plasmids using the mMessage mMachine SP6 Transcription Kit (Thermo Fisher Scientific, AM1340). RNAs were purified using NucAway Spin Columns or the Monarch RNA Clean Up Kit, following the manufacturer's instructions.

For expression of dUAS:H2B-RFP-EB3-GFP (to label polymerizing plus ends of microtubules and nuclei) the plasmid was co-injected with *gal4* RNA. For expression confined to the yolk cell, the FP2 plasmid was used as previously described (Fei et al., 2019; Narayanan and Lekven, 2012). Genes of interest were cloned from pCS2+ (Turner and Weintraub, 1994) into pFP2 using the Gibson assembly method (Gibson et al., 2009) with the Gibson Assembly Cloning Kit (NEB, E5510S). For yolk-specific expression of full-length Camsap2a, the plasmid pzTol2[Exp]-{wnt8 promoter}>{camsap2a} containing the YSL-specific *wnt8* promoter and full-length coding region of Camsap2a was generated by VectorBuilder (see Tables S1 and S2 for additional details).

## Generation of Rab5ab constructs

For sequencing, *rab5ab* was PCR amplified using Q5 High Fidelity DNA polymerase (NEB, M0491S) from 1 dpf cDNA and cloned into pGEM-T easy using the following primers: forward primer, 5′-ATGGCAGGAAGAGGTGGAGCA-3′; reverse primer, 5′-TTAGGTG-CTACAGCAGGGGC-3′. Gateway cloning (Thermo Fisher Scientific; LR clonase II, 1179120, BP clonase II, 1178920) was used to generate Rab5ab and N-terminally GFP-tagged Rab5ab expression constructs following the manufacturer's instructions and confirmed by sequencing. To generate

pDEST2:rab5ab (N-terminal entry), pGEM-T easy *rab5ab* was used as the template and the following primers were used: Rab5ab forward primer (with Kozak sequence), 5′-GGGGACAAGTTTGTACAAAAAAGCAGGCTT-CGCCACCATGGCAGGAAGAGGTGGA-3′; rab5ab reverse primer, 5′-CTTATCATGTCTGGATCATCATCCCACTTTGTACAAGAAAGCTGG-3′. Once *rab5ab* PCR with attB sites was amplified, the BP reaction was set up using attB *rab5ab* PCR, BP clonase II and pDONOR221. This was followed by LR reaction using successful entry plasmid pDONOR221-rab5ab, LR clonase II and pCSeGFPDest or pDEST2 (for untagged).

To generate pCSeGFPrab5ab (N-terminally GFP tagged Rab5ab), pGEM-T easy *rab5ab* was used as the template and the following primers were used: eGFP rab5ab forward primer, 5′-GGTCACTCACGCAAC-ACCGCCATGGCAGG-3′; rab5ab reverse primer, 5′-CTTATCATGTC-TGGATCATCATCCCACTTTGTACAAGAAAGCTGG-3′.

Site-directed mutagenesis was used to make constitutively active Rab5ab (CA-Rab5ab) by introducing a Q→L mutation, as described (Zhang et al., 2007). Using pDEST2:rab5ab as the template, the following primers were used: CA-rab5ab forward primer, 5′-ACAGCTGGCCTGGAGCGCTAC-CAC-3′; CA-rab5ab reverse primer, 5′-ATCCCAGATCTCAAACTT-CACCGT-3′.

Gibson cloning was used to clone pDEST2:rab5ab, pDEST:CA-rab5ab and pCSeGFPDest:rab5ab into the FP2 plasmid for yolk-specific expression using the following primers: forward primer, 5′-GGTCACTCACG-CAACGCCACCATGGTGAGC-3′; reverse primer, 5′ CTTATCATGTCTG-GATCATCATACCACTTTGTACAAGAAAGCTGGG-3′. Plasmid details are provided in Table S2.

### Embryo microinjection
Fertilized embryos were kept in fish facility water in a 28-29°C incubator after collection. Embryo microinjections were performed at the one-cell stage as described (Bruce et al., 2003). Injected embryos were kept in the incubator until they reached the desired stage for imaging. One-cell-stage embryos were injected with volumes ranging from 500 to 900 pl. See Tables S1 and S2 for further details.

### MZ*camsap2a* epiboly assessment
Bright-field images of live wild-type and MZ*camsap2a* mutant embryos positioned laterally at desired stages were taken on a Leica MZ16F stereomicroscope equipped with a QImaging Micropublisher 3.3 camera and Volocity 6.3 software (PerkinElmer). Images were further processed using Volocity and ImageJ. The shape of the embryo was outlined using ImageJ, and the measure tool was used to calculate the circularity of the embryos.

Epiboly progression was assessed by performing *in situ* hybridization for *tbxta* on wild-type and time-matched MZ*camsap2a* mutant embryos at 7.5 hpf, as well as on wild-type and mutant embryos expressing full-length Camsap2a or CA-Rab5ab in the yolk cell. *In situ* hybridization for *tbxta* was performed according to the protocol above. Stained embryos were imaged laterally using a Leica MZ16F stereomicroscope as above. Epiboly progression was quantified by measuring the distance from the animal pole to the margin, indicated by *tbxta* staining and divided by the total length of the embryo measured from the animal pole to the vegetal pole using ImageJ (see Fig. S1F) (West et al., 2017).

### Phalloidin staining and quantification
Embryos were manually dechorionated at desired stages and fixed in 4% paraformaldehyde (Ted Pella, 18505) overnight at 4°C. Embryos were washed the next day in 1×PBT for four 5 min washes followed by permeabilization in 0.5% Triton X-100 (Millipore Sigma, X100) in 1×PBS for 1 h. Rhodamine-phalloidin (Thermo Fisher Scientific, R415) was diluted 1:2000 in 1×PBT. Embryos were incubated in rhodamine-phalloidin overnight at 4°C and washed in 1×PBT the next day before mounting for confocal imaging.

### Whole-mount immunohistochemistry
Mouse anti-alpha tubulin (T6199, Millipore Sigma; RRID:AB 477583) was used at 1:500 and goat-anti mouse IgG highly cross-absorbed Alexa Fluor Plus 488 secondary antibodies (A32723, Invitrogen) were used at

1:1000. Microtubule antibody staining was performed as previously described (Topczewski and Solnica-Krezel, 1999) with the following modification: embryos were fixed for 2 h at room temperature in 3.7% formaldehyde (Thermo Scientific, 28908), 0.25% glutaraldehyde (Ted Pella, 18420), 0.2% Triton X-100 (Millipore Sigma) in general tubulin buffer (Cytoskeleton, BST01).

### Imaging platforms and image acquisition
Confocal time-lapse movies and images were captured on a Leica TCS SP8 confocal microscope using 25×, 40× and 60× objectives. Time-lapse movies at high temporal resolution were acquired using a Nikon ECLIPSE Ti2 spinning disk confocal microscope and a 60× (1.4 NA) objective with a rate of 500 ms/frame. To image fixed samples for quantification purposes, samples were imaged at the same stack size using the same laser power and objective to ensure consistency of the imaging data. Manually dechorionated embryos were mounted in 0.8% LMA on glass-bottom dishes (MatTek, P35G-1.0-14-C).

### Image analysis and quantification
#### Microtubule polymerization rate
Microtubule plus-end growth was visualized by expressing Gal4/dUAS: H2B-RFP-EB3-GFP. EB3 dynamics were captured using a Nikon ECLIPSE Ti2 spinning disk confocal microscope with a 60× objective on a single plane. EB3 comets in the e-YSL and yolk cytoplasmic layer were distinguished by their location and direction of movement and were cropped to the same region size for analysis. U-track software in MATLAB was used to calculate the average speed of EB3 comets in the regions of interest (Applegate et al., 2011).

#### Actin fluorescence intensity
For phalloidin fluorescence intensity (Fig. 3), time-matched and stage-matched wild-type and mutant embryos were imaged on the same day using the same laser power and stack size on a Leica TCS SP8 confocal microscope. To compare e-YSL levels in wild-type and mutant time-matched embryos, e-YSL actin intensity was measured and compared. To compare mutant time-matched and stage-matched embryos, phalloidin fluorescence intensity was normalized to each embryo's blastoderm.

#### Actin flow tracking and PIV analysis
Live confocal time-lapse movies were taken on a Nikon ECLIPSE Ti2 spinning disk confocal microscope using a 60× (1.4 N.A.) objective for 3-5 min at a rate of 500 ms/frame. A stack size of ~2.7 μm, consisting of all cortical yolk actin networks, was used for analysis. Time-lapse movies were cropped to limit the analysis to the yolk actin only and cropped regions of interests had the same width of 50 μm, while the length of the region varied depending on the amount of the yolk cell captured in the images. Adjusted time-lapse movies were oriented with the EVL-YSL margin on the top and the vegetal end of the yolk cell at the bottom to generate directionality profiles for each of the movies, and they were saved as tiff files to be analyzed. To quantify the flow of actin, PIV was performed. In brief, the open-source library (openpiv) was used for the PIV. This algorithm tracks the motion of the seeding particles (which are the actin puncta in this case) and calculates speed and direction (the velocity field) of the particle flow. It compares the actin puncta within a region of interest (~14.8×14.8 μm) on an image to the same region of interest in the subsequent frame. Based on the flow detected, the algorithm generates a cross-correlation map. The displacement of actin puncta was measured by finding the maximum coefficient within the resulting cross-correlation map. To filter spurious vectors, only vectors that had a cross-correlation coefficient above a threshold of 1 were kept. Furthermore, we removed abnormally fast vectors with flow velocity above 1 μm/min based on the range of actin flow reported in other studies (Behrndt et al., 2012). The complete algorithm for this analysis was implemented using a custom-made Python script available in Github (https://github.com/ernestiu/actin-flow-PIV-code.git).

### FRAP analysis
FRAP experiments on rhodamine-actin injected wild-type and mutant embryos were performed on a Nikon ECLIPSE Ti2 spinning disk confocal

microscope using a 60× objective (1.4 N.A). The size of the photobleached region at the margin was consistent across all embryos within one experiment. Photobleaching of rhodamine-actin was performed using the 561 nm laser under the following setting: 100% laser power, 2000 Hz ablation frequency, 100 μs dwell time and 300 ms stimulation time. Each embryo was imaged for 5 s before stimulation, and recovery post-bleaching was recorded for 1 min. Owing to robust actin labeling using rhodamine-actin and weak laser strength, regions of interest were challenging to fully bleach. Thus, not all the fluorescence signal was depleted. Fluorescence signal recovery was measured using ImageJ and normalized using the ImageJ plugin developed by Jay Unruh from the Stowers Institute for Medical Research, MO, USA.

### Myosin accumulation measurement

To measure changes in yolk myosin accumulation in MZ*camsap2a* mutants, live confocal movies of *Tg(actb2:myl12.1-GFP)*, *Tg(actb2:myl12.1-GFP); MZcamsap2a^{uot20/uot20}* and *Tg(actb2:myl12.1-GFP);MZcamsap2a^{uot20/+}* embryos were acquired using the same laser power and stack size. Myosin-GFP fluorescence intensity in the e-YSL was measured at each recorded time frame per embryo using the measure tool in ImageJ. Yolk myosin accumulation levels were measured across the e-YSL below the margin. Marginal myosin levels were quantified by outlining the boundary between EVL and YSL and measuring the fluorescence intensity. In each experiment, myosin-GFP fluorescence intensity at the margin and below the margin across time were normalized to the fluorescence intensity measured in the initial time frame. The slope of changes in myosin-GFP fluorescence intensity over time were calculated using Excel.

### UV laser ablation

Tension in the e-YSL was assessed using UV laser ablation. In both wild-type and MZ*camsap2a* mutant embryos, UV laser ablation was performed on Nikon ECLIPSE Ti2 spinning disk confocal microscope using a 60× objective. Horizontal cuts were made in the e-YSL parallel to the margin. The size of ablated region was consistent within each experiment, ranging from a width of 2-3 μm. The ablation was performed using the 355 nm laser with a 1% laser power setting 500 Hz ablation frequency, 100 μs dwell time and 10 ms stimulation time. Embryos were imaged up to 1 min after ablation to monitor recovery.

### Membrane internalization assessment

To assess membrane internalization capability in wild-type and mutant embryos, embryos were dechorionated at the desired stage and soaked in 2 mg/ml FITC-Dextran (Thermo Fisher Scientific, D1820) for 10 min in dark. After incubation, embryos were washed in fresh facility water four times and fixed in 4% paraformaldehyde overnight at 4°C. The next day embryos were washed and stained with phalloidin (see above). Embryos were imaged with a 25× objective using the same laser power and stack size on a Leica TCS SP8 confocal microscope. Quantification of endocytosed vesicles was performed by thresholding the stack images until only endocytosed vesicles were masked. The 'Analyze Particles' tool in ImageJ was used to count the number of vesicles and record the size of each particle. The particle size was set from 0.2-2 μm to eliminate background noise from thresholding. Masked particles after the analysis were compared to the original image to ensure all endocytosed vesicles were captured. The number of vesicles counted in ImageJ were plotted using Prism software and Welch's two-tailed unpaired *t*-tests were performed to analyze the statistical significance of changes in the number of vesicles observed in wild-type and MZ*camsap2a* mutant embryos.

### Scanning electron microscopy

Wild-type and MZ*camsap2a* mutant embryos from both alleles were fixed in 2.5% glutaraldehyde (Electron Microscopy Sciences) in 0.1 M Sorenson's phosphate buffer (pH 7.4) at 4°C for several days. Embryos were then rinsed in 0.1 M Sorenson's phosphate buffer (3×10 min) and post-fixed in 1% OsO$_4$ (Electron Microscopy Sciences) for 1 h. Embryos were rinsed and dehydrated through an ascending ethanol series for 10 min for each step over 1 h. They were infiltrated with an ascending series of ethanol: hexamethyldisilazane mixture for 10 min each step over one 1 h followed by three changes in 100% hexamethyldisilazane. Samples were left to dry overnight in the fume hood and mounted the following day, sputter coated with Gold-Palladium and examined with a Hitachi SU3500 Scanning Electron Microscope. All chemicals were purchased from Sigma-Aldrich unless otherwise noted.

### Photoshop

Figures were assembled in Adobe Photoshop 2026 (v27.2.0). Minor brightness adjustments were made in some cases for display purposes only. For whole-mount *in situ* hybridization for the Camsap2a rescue experiments shown in Fig. 8A, curves and levels were used for display purposes only to make the signal more visible over the opaque yolk cell.

### Acknowledgements

We thank Audrey Chong of the Imaging Facility in the Department of Cell and Systems Biology at the University of Toronto for their technical expertise in the sample preparation and the use of the Hitachi SU3500 scanning electron microscope and Henry Hong and Kenana Al Kakouni for confocal training. We thank Masa Tada for reagents, and we thank our many work-study students for excellent fish care. For helpful discussions, we thank Donna Guan, Rudi Winklbauer, Tony Harris, Vince Tropepe and Jennifer Mitchell.

### Competing interests

The authors declare no competing or financial interests.

### Author contributions

Conceptualization: H.W., A.E.E.B.; Formal analysis: H.W., S.Q., E.I., S.D.U.; Funding acquisition: S.V.P., A.E.E.B.; Investigation: H.W., S.Q., S.D.U., A.E.E.B.; Methodology: E.I.; Project administration: A.E.E.B.; Software: E.I.; Supervision: S.V.P., A.E.E.B.; Validation: H.W., S.Q., E.I., S.D.U.; Visualization: H.W., S.Q., E.I., S.D.U.; Writing – original draft: H.W., A.E.E.B.; Writing – review & editing: H.W., S.Q., E.I., S.D.U., S.V.P., A.E.E.B.

### Funding

This work was supported by Natural Sciences and Engineering Research Council of Canada grants (458019 to A.E.E.B., 201009 to S.V.P.). E.I. was supported by an Ontario Graduate Scholarship from the Government of Ontario, and a Natural Sciences and Engineering Research Council of Canada PGS-D. Open Access funding provided by the University of Toronto. Deposited in PMC for immediate release.

### Data and resource availability

PIV analysis custom script available in Github (https://github.com/ernestiu/actin-flow-PIV-code.git). All other relevant data and details of resources can be found within the article and its supplementary information.

### Peer review history

The peer review history is available online at https://journals.biologists.com/dev/lookup/doi/10.1242/dev.204843.reviewer-comments.pdf

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
