## [Peer Review File · Development (Cambridge, England)]

Camsap2a regulates actomyosin flow and Rab5ab-mediated macropinocytosis in the yolk cell during zebrafish epiboly

Haoyu Wan, Sifa Quibria, Ernest lu, Sirma Damla User, Sergey V. Plotnikov and Ashley E. E. Bruce

DOI: 10.1242/dev.204843

Editor: Steve Wilson

Review timeline

Original submission:	8 April 2025
Editorial decision:	23 May 2025
First revision received:	6 October 2025
Editorial decision:	6 November 2025
Second revision received:	29 December 2025
Accepted:	31 December 2025

Original submission

First decision letter

MS ID#: dev.204843

MS Title: Camsap2a regulates actomyosin flow and Rab5ab-mediated macropinocytosis in the yolk cell during zebrafish epiboly

Authors: Haoyu Wan; Sifa Quibria; Ernest lu; Sergey V. Plotnikov; Ashley E. E. Bruce

Article Type: Research Article

Dear Ashley,

I have now received all the referees' reports on the above manuscript, and have reached a decision. The referees' comments are appended below, or you can access them online: please go to:

As you will see, the referees express considerable interest in your work, but have some significant criticisms and suggestions for improving your manuscript. If you are able to revise the manuscript along the lines suggested, I will be happy receive a revised version of the manuscript. Please also note that Development will normally permit only one round of major revision. If it would be helpful, you are welcome to contact us to discuss your revision in greater detail. Please send us a point-by-point response indicating your plans for addressing the referees' comments, and we will look over this and provide further guidance.

Please attend to all of the reviewers' comments and ensure that you clearly highlight all changes made in the revised manuscript. Please avoid using 'Tracked changes' in Word files as these are lost in PDF conversion. I should be grateful if you would also provide a point-by-point response detailing how you have dealt with the points raised by the reviewers in the 'Response to Reviewers' box. If you do not agree with any of their criticisms or suggestions please explain clearly why this is so.

Reviewer 1

SUMMARY OF THE ADVANCE MADE IN THIS PAPER AND ITS POTENTIAL SIGNIFICANCE TO THE FIELD

Wan et al present an elegant and thorough analysis of the effect of mutation of Camsap2a on zebrafish epiboly. This analysis reveals an unexpected effect of this microtubule associated protein on actin microfilament dynamics. It provides evidence for a connection between the roles of these different cytoskeleton components, both of which have distinctive and dynamic regional organization within the zebrafish yolk cell. In their investigation, the authors also discovered unanticipated regionalization within the marginal microfilament band, pointing to additional undescribed complexity in the cytoskeleton of the yolk cell.

SUGGESTIONS TO AUTHORS

The experiments are nearly comprehensive. However, I have a few outstanding questions. To ensure the phenotype is due to *camsap2a* mutation, does injection of *Camsap2a* mRNA rescue the epiboly delay? To confirm that the function of *camsap2a* is required in the yolk cell, does injection of *Camsap2a* mRNA into the YSL rescue the phenotype? This may be outside of the scope of this manuscript but the phenotype is mild and may be compensated by mechanisms that are not robust to temperature. Is the epiboly delay phenotype temperature sensitive?

The data are well presented although the panels could use some attention to alignment and matting. Also I recommend the following to make the work more accessible and clear:

Figure 1 or Figure 2 would benefit from the addition of a schematic showing the anatomy of the margin during epiboly and the cytoskeletal organization of the yolk cell. This would make this paper accessible to a wider audience.

In Figure 3 panels D, D', E and E' are a bit confusing. The abbreviations TM and SM are not defined in the legend. Presumably, these are Time Matched and Stage Matched? But the legend states the D&E are Time matched and D'&E' are Stage Matched but the upper panels show a different calculation than the lower.

In Figure 4, panels A-C can the regions used for quantification be indicated?. In panels D-E', it would be helpful if graphs conducting the same measurements had the same y-axis range to allow for comparison between conditions.

In Figure 5, please change the color spectrum of the red and green fluorescence to be more color blind friendly. To demonstrate the deep actin structure, it would be helpful to show a computed xz-orthogonal slice here rather than leaving it until later or the movies.

Supplementary figure 2 A-C it is difficult to see the colocalization. Single color panels would be helpful to see the red under the green. Also, please use a different color scheme for the 2 channel panels to make them accessible to color blind readers.

It would be helpful to include stills from the supplementary movies in the main figures with annotations of the described dynamic events. Alternatively, annotations can be added to the movies themselves, perhaps in a second clip following the first unannotated clip.

Reviewer 2

SUMMARY OF THE ADVANCE MADE IN THIS PAPER AND ITS POTENTIAL SIGNIFICANCE TO THE FIELD

Epiboly is a key morphogenetic process that spreads the embryonic blastoderm over the yolk, enabling subsequent development during and after gastrulation. Here, Wan et al characterizes a role for *Camsap2a*, a microtubule (MT) stabilizing protein, in epiboly progression in zebrafish. They generate 2 presumed loss-of-function mutations in *camsap2a*, finding that these embryos can survive to fertile adults. While maternal-zygotic (MZ) *camsap2a* mutants also survive, they exhibit significant delays in epiboly during mid- to late gastrulation. The authors claim that although *Camsap2a* is described as a MT interactor, these mutants exhibit no apparent MT phenotypes and instead show reduced actomyosin accumulation and abnormal actin flow within the yolk. Laser cutting measurements suggest that this results in reduced tension from reduced contractility of the

contractile actomyosin ring. Previous work showed that macropinocytosis removes yolk membrane from in front of the epibolizing blastoderm, and the authors show here that the number of internalized vesicles is reduced in the yolks of *camsap2a* mutants. They go on to describe a population of deep actin puncta within the yolk that they claim are closely associated with these vesicles. They further show that *camsap2a* mutants exhibit enlarged membrane accumulation near these deep actin puncta, consistent with increased macropinocytosis. This seems at odds, however, with the previous claim that macropinocytosis is reduced in these mutants. Finally, the authors demonstrate that constitutively active Rab5 restores the number of internalized vesicles, actin accumulation, and epiboly progression in *camsap2a* mutants. Together, these data partially support a model in which Camsap2a activates Rab5 to promote macropinocytosis of the yolk, which promotes actin remodeling at the YSL actomyosin ring to drive epiboly. However, additional data are required to fully support this model, and to validate some tools and approaches presented, before the study would be appropriate for publication. Specific recommendations are provided below:

SUGGESTIONS TO AUTHORS

1. The authors describe their mutants as probable loss of function alleles, but provide no validation. Does the *uot20* mutation induce nonsense-mediated decay, for example?
2. The authors hypothesize that *camsap2a* functions within the yolk to promote epiboly... does injection of *camsap2a* RNA into the YSL rescue the epiboly delay in mutant embryos, as would be predicted by this hypothesis? Does injection of *camsap2a* RNA at one-cell stage rescue the mutants?
3. The authors claim that MT organization appears normal in *camsap2a* mutants, but can this be quantified in some way? WTs and mutants appear different in terms of MT abundance and orientation to this reviewer.
4. They similarly show PIV analysis of actin flows in mutant embryos, but no statistics are provided in support of their claims of altered direction of flow. In Figure, 3, it is not clear what the difference between 3l' and 3l" is? Are these different populations of WTs? If so, A) how many individual embryos are represented in each graph? and B) the 2 different WT populations appear more different from one another than between WTs mutants, which suggests that differences may be more due to inter-clutch variability than to loss of *camsap2a*. It would also be good to analyze the speed of actin flow, which looks difference by PIV.
5. Figure 4 shows reduced Myosin accumulation in *camsap2a* mutant embryos, but it is unclear how and from where these measurements were made. It would be helpful to include a diagram of which region of the embryos were measured. Also, Myosin levels appear dramatically reduced in mutant embryos even at the first timepoint, which is not reflected in the quantifications normalized to the first timepoint. Perhaps a different way of quantifying and graphing the data would better reflect this difference?
6. In several instances, no embryo images are presented in support of quantification shown. For example, readers need to see the laser cutting experiments in Figure 4 and all of the Rab5 phenotype suppression presented in Figure 6.
7. For the laser cutting experiments in Figure 4, the recoil velocities measured in the WT groups with each of the mutant alleles are very different. For both horizontal and vertical cuts, the WT group for *uot20* resembles the *uot19* mutants. Why such dramatic differences between WT groups, and how then can the authors be confident in their comparisons with mutants? Also, only 2 data points are shown for vertical WT controls for *uot20*, which seems too small for statistical significance. Is each datapoint a single cut from a single embryo?
8. For the dextran internalization studies in Figure 5, some negative controls should be performed. For example, showing a lack of puncta in the absence of the dextran, and an inhibitor of endocytosis/macropinocytosis.
9. I am very confused about what the deep actin puncta are supposed to represent. The authors claim they are associated with internalized vesicles, although the data in Supp Fig. 2 are

not convincing of this, particularly because only a single overlay image is presented. They are also supposed to be associated with internalized membrane, which is increased/enlarged in mutants compared with WTs. First, these data are also not convincingly presented, and second, this seems contradictory to the data and claims supporting a reduction of macropinocytosis in mutants, whereas this seems to show the opposite.

10. I feel the inclusion of the Arp2/3 inhibitor adds little to the story as presented. Especially because no evidence is provided to validate the efficacy of this inhibitor.

11. The fact that WT-Rab5 doesn't suppress macropinocytosis phenotypes in *camsap2a* mutants while CA-Rab5 does is among the most convincing and exciting pieces of data in this study. However, WT-Rab5 was not tested for its ability to restore normal actin levels and epiboly progression. These experiments will be important to establish a role for Rab5 downstream of *Camsap2a* in these processes.

12. The authors write that they hypothesize expression of Rab5 will rescue the *camsap2a* mutants, but there is not evidence that *rab5* expression is reduced. Indeed, the data presented support a role for *Camsap2a* in activating Rab5, so language discussing expression should be removed and replaced with language about activation.

13. The authors should avoid the use of red-green color schemes in figure to ensure accessibility to colorblind readers.

Reviewer 3

SUMMARY OF THE ADVANCE MADE IN THIS PAPER AND ITS POTENTIAL SIGNIFICANCE TO THE FIELD

This interesting manuscript reports studies of *Camsap2a*, a non-centrosomal microtubule stabilizing protein, in zebrafish development, implicating it in the process of epiboly progression. Utilizing two distinct *camsap2a* mutant alleles they demonstrate defects in epiboly progression in maternal-zygotic mutants. By *in vivo* analyses of the yolk microtubule and actin cytoskeleton, they conclude that the transient epiboly defects in the mutant of *Camsap2a* are largely due to defects in actin flow and formation and function of the contractile band in the yolk cell previously shown to be the main motor of the epiboly process. They also show the requirement of *Camsap2a* in the process of micropinocytosis required for removing yolk cell membrane in front of the advancing blastoderm. Finally, they place *Camsap2a* upstream of the small GTPase Rab5ab in this process. The work is generally well documented and clearly presented. The significance is in advancing our understanding of the molecular and cellular processes of one of the largest *in vivo* microtubule-actomyosin process, which is the zebrafish epiboly. This work helps to understand the molecular requirements for the actin flow in the yolk cell that builds the actomyosin ring and for the micropinocytosis process, and how the two maybe interconnected. The second advance is providing experimental evidence that *Camsap2a*, known as a non-centrosomal microtubule stabilizing protein, to be primarily required for actin cytoskeleton organization, dynamics and function. Hence, this work should be of interest to both developmental and cell biologists.

SUGGESTIONS TO AUTHORS

There are several questions about the presented results and some conclusions will require additional experimental support for the manuscript to be suitable for publication.

Main points:

1. The two generated *camsap2a* mutant alleles, *uot19* and *uot20* should be more thoroughly characterized. What is transcript level in MZ mutant embryos at maternal and zygotic stages?
2. Given that *camsap2a* is one of five *camsap* zebrafish homologs the possibility of transcriptional adaptation that could occur in *uot20* nonsense mutant allele should be evaluated, especially if nonsense-mediated degradation occurs (#1 above). This is important as no significant effect on microtubule cytoskeleton is observed in either mutant and this could be due to upregulation of other homologs.

3. MZcampsap2a epiboly phenotype is described. It will be important to present quantitative assessment of epiboly progression in zygotic mutants and also strict maternal mutants to understand the contributions of both zygotic and maternal campsap2a gene expression.
4. The conclusion that "yolk cell microtubule organization appeared normal is based on in vivo imaging experiments with embryos injected with the microtubule binding domain of ensconsin EMTB-3XGFP. As the authors note, they observed some variability in fluorescence intensity between embryos due to differential distribution of the injected synthetic transcript. However, some differences between mutant and wild-type embryos are observed. Therefore, analysis with an orthogonal method would be important, for example anti-tubulin antibody stainings. Not essential, but could be considered, testing sensitivity of the mutants to microtubule stabilizing or destabilizing agents.
5. The actin accumulation in EVL cells is reduced and EVL cell shapes are altered in the mutants compared to wild type. The authors think that this phenotype "is secondary to the reduction in pulling force from the yolk actomyosin ring (Marsal et al., 2021)". If so, this phenotype along with the overall epiboly phenotype should be rescued by expressing campsap2a in the yolk cell or in the experiments with carab5a suppression. This was not clearly stated. Alternatively, transplanting mutant cells to wild-type hosts could further test if this is cell autonomous or non-autonomous EVL phenotype.
6. Some aspects of the mutant phenotype, like actin.... are improved at late epiboly, but other like defective membrane ruffles (beautiful images) not. Can the authors comment on this?
7. Whereas the model of Campsap2a acting upstream and regulating Rab5ab is attractive and supported by the presented experimental observations, additional evidence is needed. Only one image of GFP-Rabtab is shown along with Rhodamine-Actin in wild type. It would be important to ask how Rab5ab distribution is altered in mutant embryos, and do the two proteins colocalize during epiboly?

Additional considerations:

1. Figures are beautiful but they could be more reader-friendly and informative if the labels showing the transgenes, RNA etc were provided on the figures.
2. In Figure 4 developmental time is indicated by epiboly/shield stages while in plots (D,F) in hpf. Please unify.
3. "wild type" as a noun while "wild-type embryo" as an adjective. Please correct.
4. Lines 226-227 "From 7 hpf onwards there was a significant reduction in the number internalized vesicles was observed for both mutant alleles," Please rephrase

First revision

Author response to reviewers' comments

We appreciate the helpful comments from the reviewers. Please find our responses below:

Reviewer 1: SUGGESTIONS TO AUTHORS

The experiments are nearly comprehensive. However, I have a few outstanding questions. To ensure the phenotype is due to campsap2a mutation, does injection of Campsap2a mRNA rescue the epiboly delay? To confirm that the function of campsap2a is required in the yolk cell, does injection of Campsap2a mRNA into the YSL rescue the phenotype? This may be outside of the scope of this manuscript but the phenotype is mild and may be compensated by mechanisms that are not robust to temperature. Is the epiboly delay phenotype temperature sensitive?

1. Response: We now include data showing that we can rescue the epiboly delay, actin accumulation and macropinocytosis in mutant embryos by expressing Campsap2a exclusively in the yolk cell (Figure 8). While the potential temperature sensitive nature of the mutation is an interesting point, it is beyond the scope of this work.

The data are well presented although the panels could use some attention to alignment and matting. Also I recommend the following to make the work more accessible and clear:

Figure 1 or Figure 2 would benefit from the addition of a schematic showing the anatomy of the margin during epiboly and the cytoskeletal organization of the yolk cell. This would make this paper accessible to a wider audience.

2. Response: We now include a schematic in Figure S1C. The microtubules are not shown but they can be seen in the new antibody staining images in Figure 2A.

In Figure 3 panels D, D', E and E' are a bit confusing. The abbreviations TM and SM are not defined in the legend. Presumably, these are Time Matched and Stage Matched? But the legend states the D&E are Time matched and D'&E' are Stage Matched but the upper panels show a different calculation than the lower.

3. Response: We apologize for the confusion and have revised this figure.

In Figure 4, panels A-C can the regions used for quantification be indicated? In panels D-E', it would be helpful if graphs conducting the same measurements had the same y-axis range to allow for comparison between conditions.

4. Response: This is now Figure 5. The Y-axis has been standardized and regions quantified indicated in a schematic in Fig. S1C.

In Figure 5, please change the color spectrum of the red and green fluorescence to be more color blind friendly. To demonstrate the deep actin structure, it would be helpful to show a computed xz-orthogonal slice here rather than leaving it until later or the movies.

5. Response: This is now Figure 7. The colors have been changed, and the deep actin puncta data have been removed (see below).

Supplementary figure 2 A-C it is difficult to see the colocalization. Single color panels would be helpful to see the red under the green. Also, please use a different color scheme for the 2 channel panels to make them accessible to color blind readers.

6. Response: We agree that these images are difficult to see and given the need for more experiments and better image quality, we have chosen to remove it. Removal of this data does not impact the conclusions of the work.

It would be helpful to include stills from the supplementary movies in the main figures with annotations of the described dynamic events. Alternatively, annotations can be added to the movies themselves, perhaps in a second clip following the first unannotated clip.

7. Response: See above—these movies have been removed.

Reviewer 2: SUGGESTIONS TO AUTHORS

1. The authors describe their mutants as probable loss of function alleles but provide no validation. Does the *uot20* mutation induce nonsense-mediated decay, for example?

Response: qPCR data for all 5 *camsap* genes is now included (Figure S1F). We see upregulation of other *camsaps* though interestingly we do not see significant down regulation of *camsap2a* itself. We discuss the possibilities that the transcript is upregulated in mutants, and/or that the nonsense mediated decay machinery might be different in the yolk cell. We agree that we are unable to verify that *uot20* is a null allele and we have tempered the text accordingly. At this time, we have no way to examine the expression of the endogenous protein.

2. The authors hypothesize that *camsap2a* functions within the yolk to promote epiboly... does injection of *camsap2a* RNA into the YSL rescue the epiboly delay in mutant embryos, as would be predicted by this hypothesis? Does injection of *camsap2a* RNA at one-cell stage rescue the mutants?

Response: See response 1 to reviewer 1 above. Given the yolk localized expression of *camsap2a*, we now present yolk specific rescue, and we find that all aspects of the phenotype are rescued (Figure 8).

3. The authors claim that MT organization appears normal in *camsap2a* mutants, but can this be quantified in some way? WTs and mutants appear different in terms of MT abundance and orientation to this reviewer.

Response: While it is not possible to quantify the yolk cell microtubules due to technical challenges of antibody penetration and imaging, we now include tubulin antibody staining which demonstrates no obvious differences between wild type and mutants (Fig. 2A). We also added stills from time-lapse imaging of labeled microtubules to present a more complete picture of the microtubules over time in wild type and mutants. We discuss the fact that we cannot rule out some role for microtubules but given that it is well established that yolk actomyosin is the main driver of epiboly, we focus our analyses there.

4. They similarly show PIV analysis of actin flows in mutant embryos, but no statistics are provided in support of their claims of altered direction of flow. In Figure, 3, it is not clear what the difference between 3l' and 3l'' is? Are these different populations of WTs? If so, A) how many individual embryos are represented in each graph? and B) the 2 different WT populations appear more different from one another than between WTs mutants, which suggests that differences may be more due to inter-clutch variability than to loss of *camsap2a*. It would also be good to analyze the speed of actin flow, which looks difference by PIV.

Response: This figure has been substantially revised and re-analyzed (it is now Figure 4). The numbers of embryos, stats and detailed Methods have been included and updated. An analysis of actin flow speed has also been added (Fig. 4E).

5. Figure 4 shows reduced Myosin accumulation in *camsap2a* mutant embryos, but it is unclear how and from where these measurements were made. It would be helpful to include a diagram of which region of the embryos were measured. Also, Myosin levels appear dramatically reduced in mutant embryos even at the first timepoint, which is not reflected in the quantifications normalized to the first timepoint. Perhaps a different way of quantifying and graphing the data would better reflect this difference?

Response: We added a schematic to indicate the e-YSL region used to assess fluorescence intensity (Fig. S1C). We have clarified in the text that myosin accumulation over-time was used because of potential differences in transgene number which means that although the myosin levels may be reduced in the mutant embryos, we cannot conclude that from this data. Instead, we examined the rate of myosin accumulation in the e-YSL as an indicator of myosin flow, which we hope is now clearer in the text.

6. In several instances, no embryo images are presented in support of quantification shown. For example, readers need to see the laser cutting experiments in Figure 4 and all of the Rab5 phenotype suppression presented in Figure 6.

Response: Embryo images have now been added for the laser cutting (Figure 6), *Camsap2a* rescue (Figure 8) and Rab5ab experiments (Figure 9).

7. For the laser cutting experiments in Figure 4, the recoil velocities measured in the WT groups with each of the mutant alleles are very different. For both horizontal and vertical cuts, the WT group for *uot20* resembles the *uot19* mutants. Why such dramatic differences between WT groups, and how then can the authors be confident in their comparisons with mutants? Also, only 2 data points are shown for vertical WT controls for *uot20*, which seems too small for statistical significance. Is each datapoint a single cut from a single embryo?

Response: We found that we had made a calculation error when the data were re-analyzed. This has now been corrected; in Figure 6B it can be seen that the wild-type recoil velocities are consistent in the uot19 and uot20 experiments. We removed the vertical cut data due to the small sample size and because the horizontal cuts directly reflect upward actin flow, which is defective in the mutants. We cannot generate more vertical cut data at this time, but the removal of this data does not impact our conclusions.

8. For the dextran internalization studies in Figure 5, some negative controls should be performed. For example, showing a lack of puncta in the absence of the dextran, and an inhibitor of endocytosis/macropinocytosis.

Response: This is a well-established technique in the zebrafish embryo (since 1994), and the proposed control is not standard in the field. We respectfully suggest that a separate control is not required as one control would simply be a phalloidin stained embryo, as seen in many of our figures.

9. I am very confused about what the deep actin puncta are supposed to represent. The authors claim they are associated with internalized vesicles, although the data in Supp Fig. 2 are not convincing of this, particularly because only a single overlay image is presented. They are also supposed to be associated with internalized membrane, which is increased/enlarged in mutants compared with WTs. First, these data are also not convincingly presented, and second, this seems contradictory to the data and claims supporting a reduction of macropinocytosis in mutants, whereas this seems to show the opposite.

Response: We acknowledge that we need to do additional work to acquire more data and improved images. For these reasons we have chosen to remove this data from the manuscript which has no impact on our conclusions.

10. I feel the inclusion of the Arp2/3 inhibitor adds little to the story as presented. Especially because no evidence is provided to validate the efficacy of this inhibitor.

Response: See point 9 above—we have removed this data.

11. The fact that WT-Rab5 doesn't suppress macropinocytosis phenotypes in campsap2a mutants while CA-Rab5 does is among the most convincing and exciting pieces of data in this study. However, WT-Rab5 was not tested for its ability to restore normal actin levels and epiboly progression. These experiments will be important to establish a role for Rab5 downstream of Camsap2a in these processes.

Response: We now include data on actin accumulation for WT-Rab5ab and epiboly rescue data for CA-Rab5ab (Figure 8).

12. The authors write that they hypothesize expression of Rab5 will rescue the campsap2a mutants, but there is not evidence that rab5 expression is reduced. Indeed, the data presented support a role for Camsap2a in activating Rab5, so language discussing expression should be removed and replaced with language about activation.

Response: The text has been revised accordingly.

13. The authors should avoid the use of red-green color schemes in figure to ensure accessibility to colorblind readers.

Response: All relevant images have been revised.

Reviewer 3: SUGGESTIONS TO AUTHORS

There are several questions about the presented results and some conclusions will require additional experimental support for the manuscript to be suitable for publication.

Main points:

1. The two generated *camsap2a* mutant alleles, *uot19* and *uot20* should be more thoroughly characterized. What is transcript level in MZ mutant embryos at maternal and zygotic stages?

Response: We have now added text that our initial experiments with zygotic mutants produced embryos with mild and incompletely penetrant epiboly delays, suggesting that maternal *camsap2a* contributes to the phenotype. As our focus is not on the relative contribution of maternal and zygotic *camsap2a*, we suggest that such an analysis is outside of the scope of this work.

2. Given that *camsap2a* is one of five *camsap* zebrafish homologs the possibility of transcriptional adaptation that could occur in *uot20* nonsense mutant allele should be evaluated, especially if nonsense-mediated degradation occurs (#1 above). This is important as no significant effect on microtubule cytoskeleton is observed in either mutant and this could be due to upregulation of other homologs.

Response: We have now added a qPCR analysis (Fig. S1F and see response to reviewer 2 point 1).

3. MZ*camsap2a* epiboly phenotype is described. It will be important to present quantitative assessment of epiboly progression in zygotic mutants and also strict maternal mutants to understand the contributions of both zygotic and maternal *camsap2a* gene expression.

Response: Quantified epiboly progression data is presented in Figures 8 and 9. We have focused our analysis on MZ mutants to limit the contribution of maternal and zygotic *camsap2a* (see point 1 above). We also do not maintain heterozygous fish, which would be required for the proposed analysis and would take at least 6-months to perform. Thus, in our view this is outside of the scope of our current work and does not impact our conclusions.

4. The conclusion that "yolk cell microtubule organization appeared normal is based on in vivo imaging experiments with embryos injected with the microtubule binding domain of *ensconsin* EMTB-3XGFP. As the authors note, they observed some variability in fluorescence intensity between embryos due to differential distribution of the injected synthetic transcript. However, some differences between mutant and wild-type embryos are observed. Therefore, analysis with an orthogonal method would be important, for example anti-tubulin antibody stainings. Not essential, but could be considered, testing sensitivity of the mutants to microtubule stabilizing or destabilizing agents.

Response: See response 3 to reviewer 2. Antibody staining and additional live imaging data have been added to Figure 2. The drug treatment experiments would likely be challenging to perform and to draw meaningful conclusions from and our data strongly support the conclusion that the main defects in the mutants are actin based. We discuss that fact that there could still be a role for microtubules.

5. The actin accumulation in EVL cells is reduced and EVL cell shapes are altered in the mutants compared to wild type. The authors think that this phenotype "is secondary to the reduction in pulling force from the yolk actomyosin ring (Marsal et al., 2021)". If so, this phenotype along with the overall epiboly phenotype should be rescued by expressing *camsap2a* in the yolk cell or in the experiments with *carab5a* suppression. This was not clearly stated. Alternatively, transplanting mutant cells to wild-type hosts could further test if this is cell autonomous or non-autonomous EVL phenotype.

Response: We now show that EVL actin levels are restored after yolk cell specific expression of *camsap2a* (Figure 8).

6. Some aspects of the mutant phenotype, like actin.... are improved at late epiboly, but other like defective membrane ruffles (beautiful images) not. Can the authors comment on this?

Response: With the addition of the qPCR data, we now discuss that the viability (as a result of the eventual completion of epiboly) is likely due to the up-regulated expression of the other *camsap* genes but they cannot fully compensate for the *camsap2a* mutation.

7. Whereas the model of Camsap2a acting upstream and regulating Rab5ab is attractive and supported by the presented experimental observations, additional evidence is needed. Only one image of GFP-Rab5ab is shown along with Rhodamine-Actin in wild type. It would be important to ask how Rab5ab distribution is altered in mutant embryos, and do the two proteins colocalize during epiboly?

Response: Unfortunately, our attempts to examine endogenous Rab5 in the yolk cell using antibody staining were inconclusive due to the technical challenges of staining in the yolk cell and the tagged version may give misleading results due to over-expression artifacts. Characterizing Rab5 localization will be a goal of future work but the data we present that CA-Rab5ab rescues the Camsap2a mutant phenotype supports our model that Camsap2a plays an important role in Rab5ab activation.

Additional considerations:

1. Figures are beautiful but they could be more reader-friendly and informative if the labels showing the transgenes, RNA etc were provided on the figures.

Response: We have worked to improve the clarity of the figures.

2. In Figure 4 developmental time is indicated by epiboly/shield stages while in plots (D,F) in hpf. Please unify.

Response: This has been corrected.

3. "wild type" as a noun while "wild-type embryo" as an adjective. Please correct.

Response: We have done our best to fix this.

4. Lines 226-227 "From 7 hpf onwards there was a significant reduction in the number internalized vesicles was observed for both mutant alleles," Please rephrase.

Response: Changed to "After 7hpf, a significant reduction in the number of internalized vesicles was observed in mutant embryos, corresponding to the time when the epiboly delay was prominent (Fig. 7C,F).

Second decision letter

MS ID#: dev.204843R1

MS Title: Camsap2a regulates actomyosin flow and Rab5ab-mediated macropinocytosis in the yolk cell during zebrafish epiboly

Authors: Haoyu Wan; Sifa Quibria; Ernest lu; Sirma Damla User; Sergey V. Plotnikov; Ashley E. E. Bruce

Article Type: Research Article

Dear Ashley,

I have now received all the referees reports on the above manuscript, and have reached a decision. The referees' comments are appended below.

The overall evaluation is positive and we would like to publish a revised manuscript in Development, provided that the referees' remaining comments and suggestions can be satisfactorily addressed. Please attend to all of the reviewers' comments in your revised manuscript and detail them in your point-by-point response. Please send us a point-by-point response indicating your

plans for addressing the referees' comments, and we will look over this and provide further guidance.

Reviewer 1

SUMMARY OF THE ADVANCE MADE IN THIS PAPER AND ITS POTENTIAL SIGNIFICANCE TO THE FIELD

Wan et al have revised their manuscript and present a much improved analysis of the loss of Camsap2a. This analysis reveals an unexpected effect of this microtubule associated protein on actin microfilament dynamics. It provides evidence for a connection between the roles of these different cytoskeleton components, both of which have distinctive and dynamic regional organization within the zebrafish yolk cell. In this revision the authors have added the rescue of the mutant phenotype which solidifies their conclusions about the cause of their phenotype, and it opens the door to future structure-function experiments to dissect the roles of Camsap2a in the regulation of the cytoskeleton.

Reviewer 2

SUMMARY OF THE ADVANCE MADE IN THIS PAPER AND ITS POTENTIAL SIGNIFICANCE TO THE FIELD

SUGGESTIONS TO AUTHORS

The Authors have addressed any of my original concerns with these revisions. A few outstanding concerns are:

1. Without antibody staining to assess protein levels or evidence of nonsense-mediated decay, evidence that campsap2a function is lost in the mutant lines examined is not as strong as it could be. However, the fact that full length campsap2a expression rescues many mutant phenotypes does provide some evidence of this.
2. The Authors show by qPCR that campsap2a RNA is not downregulated in their mutants and that other campsap genes are upregulated. These data should be presented earlier in the manuscript, when the alleles and their likely effects on gene function are first discussed.
3. The Authors state that figure 5 shows that uot20/+ embryos show faster myosin accumulation than uot20 homozygotes, but the comparisons shown in the graphs are between WT and -/- or WT and +/- embryos... the stated comparison between hets and homozygotes is not shown
4. The Authors show that endocytosis and actin levels are restored in campsap2a mutant embryos by CA-, but not WT-Rab5a. They further show that epiboly progression is rescued by CA-Rab5a, but do not show data for WT-Rab5a
5. Many figures could still benefit from clearer labeling to enhance reader friendliness. For example, Fig. 8 shows the rescue of mutant phenotypes by full length campsap2a, but these embryos are labeled only as "inj". What they were injected with is not clear from the figure alone. Many graphs are also labeled simply "FI" without specifying which reporter or stain was measured

Reviewer 3

SUMMARY OF THE ADVANCE MADE IN THIS PAPER AND ITS POTENTIAL SIGNIFICANCE TO THE FIELD

This interesting manuscript reports studies of Camsap2a, a non-centrosomal microtubule stabilizing protein, in zebrafish development, implicating it in the process of epiboly progression. Utilizing two distinct campsap2a mutant alleles they demonstrate defects in epiboly progression in maternal-zygotic mutants. By in vivo analyses of the yolk microtubule and actin cytoskeleton, they conclude that the transient epiboly defects in the mutant of Camsap2a are largely due to defects in actin flow and formation and function of the contractile band in the yolk cell previously shown to be the

main motor of the epiboly process. They also show the requirement of Camsap2a in the process of micropinocytosis required for removing yolk cell membrane in front of the advancing blastoderm. Evidence is provided that Camsap2a function is required in the yolk cell, the main actor in epiboly. Finally, they place Camsap2a upstream of the small GTPase Rab5ab in this process. The work is generally well documented and clearly presented. The significance is in advancing our understanding of the molecular and cellular processes of one of the largest *in vivo* microtubule-actomyosin process, which is the zebrafish epiboly. This work helps to understand the molecular requirements for the actin flow in the yolk cell that builds the actomyosin ring and for the micropinocytosis process, and how the two maybe interconnected. The second advance is providing experimental evidence that Camsap2a, known as a non-centrosomal microtubule stabilizing protein, to be primarily required for actin cytoskeleton organization, dynamics and function. Hence, this work should be of interest to both developmental and cell biologists.

SUGGESTIONS TO AUTHORS

The authors made significant effort to address the reviewers questions and concerns by performing additional experiments (rescue by injection of campsap2a RNA into the yolk cell), removing figures that are not fully supported by the available data and toning down their conclusion when needed. This improved manuscript provides significant advance in understanding of the molecular regulation of the yolk cell actin cytoskeleton, the key driver of epiboly. Given the evolutionary conservation of Camsaps this work will be of broad interest as it provides new insights about the function of these proteins in regulation of actin and microtubule cytoskeletons. The manuscript is suitable for publication upon addressing the following small editorial points:

1. In the citation of Jesuthasan and Strähle, 1993 - the latter name is misspelled: Stähle.
2. Figure 2, please add labels to make clear that A panels show microtubules detected by immunofluorescence while the remaining panels show still images of microtubules from embryos expressing ensconsin fused to 3 GFP molecules.
3. Figure 6 would greatly benefit from additional labels either indicating mediolateral direction and/or marks indicating which dimension the openings forming upon ablation was measured. Please clarify.
4. The 4th sentence in the Discussion "Actin and myosin flow upwards from the vegetal pole at the start of epiboly and assemble into a contractile band in the during epiboly progression stages (Behrndt et al., 2012)," requires revising.

Second revision

Author response to reviewers' comments

Reviewer 1: SUMMARY OF THE ADVANCE MADE IN THIS PAPER AND ITS POTENTIAL SIGNIFICANCE TO THE FIELD

Wan et al have revised their manuscript and present a much improved analysis of the loss of Camsap2a. This analysis reveals an unexpected effect of this microtubule associated protein on actin microfilament dynamics. It provides evidence for a connection between the roles of these different cytoskeleton components, both of which have distinctive and dynamic regional organization within the zebrafish yolk cell. In this revision the authors have added the rescue of the mutant phenotype which solidifies their conclusions about the cause of their phenotype, and it opens the door to future structure-function experiments to dissect the roles of Camsap2a in the regulation of the cytoskeleton.

SUGGESTIONS TO AUTHORS

Reviewer 2: SUMMARY OF THE ADVANCE MADE IN THIS PAPER AND ITS POTENTIAL SIGNIFICANCE TO THE FIELD

SUGGESTIONS TO AUTHORS

The Authors have addressed any of my original concerns with these revisions. A few outstanding concerns are:

1. Without antibody staining to assess protein levels or evidence of nonsense-mediated decay, evidence that *campsap2a* function is lost in the mutant lines examined is not as strong as it could be. However, the fact that full length *campsap2a* expression rescues many mutant phenotypes does provide some evidence of this.

We have shown the yolk specific expression of full-length *Campsap2a* rescues all aspects of the mutant phenotype. We agree with Reviewer 1 that: “the rescue of the mutant phenotype [which] solidifies their conclusions about the cause of their phenotype.” We note that antibodies are often not available in zebrafish and rescue of mutant phenotypes with the wild-type version of the gene is standardly used.

2. The Authors show by qPCR that *campsap2a* RNA is not downregulated in their mutants and that other *campsap* genes are upregulated. These data should be presented earlier in the manuscript, when the alleles and their likely effects on gene function are first discussed.

This has been moved up to the section introducing the mutants.

3. The Authors state that figure 5 shows that *uot20/+* embryos show faster myosin accumulation than *uot20* homozygotes, but the comparisons shown in the graphs are between WT and *-/-* or WT and *+/-* embryos... the stated comparison between hets and homozygotes is not shown.

We have revised the text accordingly; the comparisons are made to the respective control embryos.

4. The Authors show that endocytosis and actin levels are restored in *campsap2a* mutant embryos by CA-, but not WT-Rab5a. They further show that epiboly progression is rescued by CA-Rab5a, but do not show data for WT-Rab5a

This data has now been added to Figure 9, yolk expression of WT-Rab5 has no effect on epiboly.

5. Many figures could still benefit from clearer labeling to enhance reader friendliness. For example, Fig. 8 shows the rescue of mutant phenotypes by full length *campsap2a*, but these embryos are labeled only as “inj”. What they were injected with is not clear from the figure alone. Many graphs are also labeled simply “FI” without specifying which reporter or stain was measured.

Additional labels have been added to the panels for several figures.

Reviewer 3: SUMMARY OF THE ADVANCE MADE IN THIS PAPER AND ITS POTENTIAL SIGNIFICANCE TO THE FIELD

This interesting manuscript reports studies of *Campsap2a*, a non-centrosomal microtubule stabilizing protein, in zebrafish development, implicating it in the process of epiboly progression. Utilizing two distinct *campsap2a* mutant alleles they demonstrate defects in epiboly progression in maternal-zygotic mutants. By in vivo analyses of the yolk microtubule and actin cytoskeleton, they conclude that the transient epiboly defects in the mutant of *Campsap2a* are largely due to defects in actin flow and formation and function of the contractile band in the yolk cell previously shown to be the main motor of the epiboly process. They also show the requirement of *Campsap2a* in the process of micropinocytosis required for removing yolk cell membrane in front of the advancing blastoderm. Evidence is provided that *Campsap2a* function is required in the yolk cell, the main actor in epiboly. Finally, they place *Campsap2a* upstream of the small GTPase Rab5ab in this process. The work is generally well documented and clearly presented. The significance is in advancing our understanding of the molecular and cellular processes of one of the largest in vivo microtubule-actomyosin process, which is the zebrafish epiboly. This work helps to understand the molecular requirements for the actin flow in the yolk cell that builds the actomyosin ring and for the micropinocytosis process, and how the two maybe interconnected. The second advance is providing experimental evidence that *Campsap2a*, known as a non-centrosomal microtubule stabilizing

protein, to be primarily required for actin cytoskeleton organization, dynamics and function. Hence, this work should be of interest to both developmental and cell biologists.

SUGGESTIONS TO AUTHORS

The authors made significant effort to address the reviewers questions and concerns by performing additional experiments (rescue by injection of campsap2a RNA into the yolk cell), removing figures that are not fully supported by the available data and toning down their conclusion when needed. This improved manuscript provides significant advance in understanding of the molecular regulation of the yolk cell actin cytoskeleton, the key driver of epiboly. Given the evolutionary conservation of Camsaps this work will be of broad interest as it provides new insights about the function of these proteins in regulation of actin and microtubule cytoskeletons. The manuscript is suitable for publication upon addressing the following small editorial points:

1. In the citation of Jesuthasan and Strähle, 1993 - the latter name is misspelled: Stähle.

Typo corrected.

2. Figure 2, please add labels to make clear that A panels show microtubules detected by immunofluorescence while the remaining panels show still images of microtubules from embryos expressing ensconsin fused to 3 GFP molecules.

We have done this, in addition to adding more labels to several other figures to improve clarity.

3. Figure 6 would greatly benefit from additional labels either indicating mediolateral direction and/or marks indicating which dimension the openings forming upon ablation was measured. Please clarify.

Arrows have been added to the WT panel to show that the recoil velocity was measured along the animal-vegetal axis and the relevant text has been revised.

4. The 4th sentence in the Discussion "Actin and myosin flow upwards from the vegetal pole at the start of epiboly and assemble into a contractile band in the during epiboly progression stages (Behrndt et al., 2012)," requires revising.

This sentence has been changed to: "Actin and myosin flow upwards from the vegetal pole and assemble into a contractile band by mid-epiboly stages."

Third decision letter

MS ID#: dev.204843R2

MS Title: Camsap2a regulates actomyosin flow and Rab5ab-mediated macropinocytosis in the yolk cell during zebrafish epiboly

Authors: Haoyu Wan; Sifa Quibria; Ernest lu; Sirma Damla User; Sergey V. Plotnikov; Ashley E. E. Bruce

Article Type: Research Article

Dear Ashley,

I am happy to tell you that your manuscript has been accepted for publication in Development, pending our standard publication integrity checks.